# Differential requirements for cyclase-associated protein (CAP) in actin-dependent processes of *Toxoplasma gondii*

Alex Hunt[1], Matthew Robert Geoffrey Russell[2†], Jeanette Wagener[1†], Robyn Kent[3], Romain Carmeille[4], Christopher J Peddie[2], Lucy Collinson[2], Aoife Heaslip[4], Gary E Ward[3], Moritz Treeck[1]*

[1]Signalling in Apicomplexan Parasites Laboratory, The Francis Crick Institute, London, United Kingdom; [2]Electron Microscopy Science Technology Platform, The Francis Crick Institute, London, United Kingdom; [3]Department of Microbiology and Molecular Genetics, University of Vermont Larner College of Medicine, Burlington, United States; [4]Department of Molecular and Cell Biology, University of Connecticut, Storrs, United States

**\*For correspondence:**
moritz.treeck@crick.ac.uk

†These authors contributed equally to this work

**Competing interests:** The authors declare that no competing interests exist.

**Abstract** *Toxoplasma gondii* contains a limited subset of actin binding proteins. Here we show that the putative actin regulator cyclase-associated protein (CAP) is present in two different isoforms and its deletion leads to significant defects in some but not all actin dependent processes. We observe defects in cell-cell communication, daughter cell orientation and the juxtanuclear accumulation of actin, but only modest defects in synchronicity of division and no defect in the replication of the apicoplast. 3D electron microscopy reveals that loss of CAP results in a defect in formation of a normal central residual body, but parasites remain connected within the vacuole. This dissociates synchronicity of division and parasite rosetting and reveals that establishment and maintenance of the residual body may be more complex than previously thought. These results highlight the different spatial requirements for F-actin regulation in *Toxoplasma* which appear to be achieved by partially overlapping functions of actin regulators.
DOI: https://doi.org/10.7554/eLife.50598.001

## Introduction

*Toxoplasma gondii* is an obligate intracellular parasite, belonging to the Apicomplexa phylum. The Apicomplexa, which also include *Plasmodium* and *Cryptosporidium* species, pose a significant global public health burden. *Toxoplasma*, specifically, is one of the most prevalent human pathogens, chronically infecting ~30% of the world's population (*Swapna and Parkinson, 2017*). While most infections are asymptomatic, in congenitally infected and immunocompromised patients, disease outcomes are often severe and potentially fatal (*Halonen and Weiss, 2013*). During acute infection of the host, the asexual tachyzoite stage of *Toxoplasma gondii* undergoes cycles of active invasion, replication and egress from host cells. This lytic cycle leads to rapid proliferation and dissemination of the parasite throughout the host (*Black and Boothroyd, 2000*). To facilitate these processes, *Toxoplasma* utilises a unique form of locomotion, called gliding motility, which relies on actin and an unconventional myosin motor (*Frénal et al., 2017a*). This motor allows the parasite to actively invade host cells, where it forms a protective parasitophorous vacuole. Parasitophorous vacuole structural integrity and stability is sustained through the parasite's release of dense granule proteins from secretory vesicles (*Heaslip et al., 2016*). Additionally, several dense granule proteins are transported

into the host cell where they co-opt or interfere with host cell functions (*Hakimi et al., 2017*). Within the parasitophorous vacuole, *Toxoplasma* begins a unique form of cell division called endodyogeny (*Sheffield and Melton, 1968*). Here, two daughter cells are synchronously assembled within the mother cell before daughter cell budding (*Delbac et al., 2001*). This initiates at the apical pole of the mother cell and once complete, the daughter cells bud from the mother cell but remain attached at their basal pole to a central residual body: a membranous compartment containing maternal remnants of organelles and cytoskeletal elements (*Muñiz-Hernández et al., 2011*). During cell division, organelles such as mitochondria and the apicoplast, a unique and essential organelle which contributes to isoprenoid synthesis (*Vaishnava and Striepen, 2006*), also divide and are distributed between the two daughter parasites. Following multiple rounds of cell division, parasites organise into a rosette-like pattern around the residual body. Formation of the characteristic rosette pattern requires actin, several myosins, and other actin-binding proteins (*Frénal et al., 2017b*; *Haase et al., 2015*; *Jacot et al., 2013*; *Periz et al., 2017*; *Tosetti et al., 2019*). It has been hypothesised that the inter-parasite connections via the residual body are not only important for parasite organisation, but also play a key role in cell-cell communication. Such communication was measured by the transfer of reporter proteins between parasites and is believed to ensure the synchronous division of parasites within a vacuole (*Frénal et al., 2017b*). The parasites continue to replicate until host cell lysis and parasite egress. Following egress, parasites migrate to and invade new host cells and the lytic cycle repeats, leading to tissue destruction (*Black and Boothroyd, 2000*). Actin plays an essential role in the parasite's lytic cycle through function of the actin cytoskeleton and actomyosin motor complex. Despite this crucial role in apicomplexan biology, there has been difficulty visualising actin filaments in apicomplexan species (*Bannister and Mitchell, 1995*; *Sahoo et al., 2006*; *Shaw and Tilney, 1999*) and it has been suggested that as much as 98% of parasite actin is monomeric (G-actin) and not incorporated into filaments (F-actin) (*Dobrowolski et al., 1997*). This, along with structural differences found in actin of Apicomplexa (*Pospich et al., 2017*), led to the hypothesis that *Toxoplasma* F-actin has reduced stability, forming abnormally short filaments that are rapidly recycled to maintain essential cellular function (*Pospich et al., 2017*; *Skillman et al., 2011*). However, recent development of the actin-chromobody has allowed for the visualisation of F-actin structures both in the parasite and extensive networks within the parasitophorous vacuole (*Periz et al., 2017*). Taken together, along with biochemical evidence, apicomplexan actin appears to be different to actin from other organisms (*Frénal et al., 2017a*).

Actin turnover is regulated by actin binding proteins, of which *Toxoplasma* possesses a reduced repertoire, including ADF, profilin, coronin and cyclase-associated protein (CAP) (*Baum et al., 2006*). Functional studies have shown ADF and profilin to be essential for *Toxoplasma* progression through the lytic cycle, while coronin depletion had a modest impact on parasite invasion and egress (*Mehta and Sibley, 2011*; *Plattner et al., 2008*; *Salamun et al., 2014*). Apart from its localisation, the function of CAP in *Toxoplasma* has not been investigated. In the majority of eukaryotes, CAP is a highly conserved multidomain protein that regulates actin filament dynamics via two distinct mechanisms (*Ono, 2013*). CAP can bind and sequester G-actin using its CAP and X-linked retinitis pigmentosa two protein (CARP) domain and can regulate actin filament disassembly by promoting ADF/cofilin mediated severing using its helical folded domain (HFD). Through regulation of actin dynamics, by interacting with actin and other actin binding proteins, it has been shown that mouse CAP1 plays important roles in cell morphology, migration and endocytosis (*Bertling et al., 2004*). However, species belonging to the Apicomplexa phylum possess a truncated form of CAP, retaining only the conserved C-terminal G-actin-binding CARP domain (*Hliscs et al., 2010*). This conserved β-sheet domain has been shown to interact directly with monomeric actin, providing either sequestration or nucleotide exchange of G-actin in a concentration-dependent manner (*Hliscs et al., 2010*; *Makkonen et al., 2013*; *Mattila et al., 2004*). As such, apicomplexan CAP is hypothesised to regulate actin turnover solely through interaction with monomeric actin. Biochemical analysis of *Cryptosporidium parvum* CAP identified the formation of a dimer and G-actin sequestering activity, while *Plasmodium falciparum* CAP was shown to facilitate nucleotide exchange, loading ADP-actin monomers with ATP (*Hliscs et al., 2010*; *Makkonen et al., 2013*). A *Plasmodium berghei* CAP KO demonstrated that while PbCAP is dispensable for asexual blood stages in vivo, there is a complete defect in oocyst development in the insect vector which was overcome through complementation with *C. parvum* CAP (*Hliscs et al., 2010*). A PbCAP overexpression study revealed no defect in ookinete motility or oocyst development, however sporozoites displayed impaired gliding motility, invasion

and salivary gland colonisation (*Sato et al., 2016*). Taken together, these results suggest that apicomplexan CAP may function as a dimer with the ability to interact with G-actin monomers to sequester them and/or facilitate their nucleotide exchange. While CAP has yet to be functionally characterised in *Toxoplasma*, Lorestani et al reported that CAP localises to the apex of intracellular parasites, a hub for events leading to egress and motility (*Graindorge et al., 2016*; *Lorestani et al., 2012*; *Tosetti et al., 2019*). Intriguingly, following host cell lysis, relocalisation of CAP to the parasite cytosol was observed (*Lorestani et al., 2012*). Furthermore, we have previously identified differential phosphorylation of CAP in parasites with a delayed egress phenotype (*Treeck et al., 2014*). The correlation between CAP redistribution and phosphorylation, following host cell lysis, hints at a potential role for CAP in actin regulation during rapid egress. Taken together, these results prompted us to characterise the role of CAP in *Toxoplasma* biology.

Here we show that *T. gondii* CAP is produced in two distinctly localised isoforms through alternative translation initiation: a membrane bound isoform, localised to the apical tip (longCAP) and a cytosolically dispersed isoform (shortCAP). Conditional knockout of CAP, using a second generation DiCre strain, identified an important function of CAP in some, but not all, actin-dependent processes. Invasion, egress, motility, correct daughter cell orientation and dense granule trafficking were all perturbed in CAP depleted parasites, while apicoplast inheritance was not. This suggests different spatial requirements for CAP in actin turnover within the cell. Furthermore, the characteristic rosette organisation of parasites in the vacuole was completely lost, but synchronicity of division was largely unaffected. Strikingly, while we observe rapid protein transfer only between two adjacent cells in a vacuole, all parasites remain connected through a decentralised residual body, potentially explaining why synchrony of division is unaffected. Using chromobody expressing parasites lines we show that CAP deletion leads to a dysregulation of actin in some, but not all subcellular localisations. In the mouse in vivo infection model, CAP-depleted type I RH parasites display normal lethality, while CAP depletion renders type II Pru parasites avirulent, with markedly reduced cyst formation. Furthermore, the cytoplasmic isoform of CAP was sufficient for the infection of mice and the formation of latent stages in the brain, indicating that the apically localised CAP isoform provides only a small fitness benefit under the conditions tested here.

## Results

### *T. gondii* CAP contains two translational start sites which results in the production of two differentially localised protein isoforms

CAP was previously shown to localise to the apex of intracellular parasites and rapidly redistribute to the cytoplasm of extracellular parasites following host cell lysis (*Lorestani et al., 2012*). This suggested that CAP localisation may be influenced by post-translational modifications that enable CAP to regulate actin dynamics at different locations in the cell. *T. gondii* CAP, and CAP from *Neospora* and *Hammondia* species contain a unique predicted N-terminal extension that is not present in other Apicomplexa, such as *Plasmodium* (*Figure 1A*). The extension contains two predicted palmitoylation sites and CAP was identified in an analysis of palmitoylated proteins in *Toxoplasma gondii* (*Foe et al., 2015*). Furthermore, two phosphorylation sites in the N-terminus of *T. gondii* CAP are substantially phosphorylated upon ionophore-induced egress (*Treeck et al., 2014*). These phosphorylation events are dependent on the calcium-dependent kinase 3 (CDPK3) (*Treeck et al., 2014*), which has been shown to be important in mediating rapid exit from the host cell and is localised to the plasma membrane (*Black et al., 2000*; *Garrison et al., 2012*; *Lourido et al., 2012*; *McCoy et al., 2012*). Collectively, these observations allow for the possibility that re-localisation of CAP is important for egress and is mediated by dynamic post-translational modifications. To visualise CAP we expressed it as a GFP fusion. CAP-GFP localises to the apex of the parasite, as previously shown, but also to the cytoplasm of intracellular tachyzoites (*Figure 1B*). We additionally demonstrated this dual localisation of CAP by C-terminally tagging the endogenous CAP locus with a HA epitope tag (*Figure 1—figure supplement 1*). To rule out any mis-localisation of the protein as a result of tagging, we expressed recombinant *T. gondii* CAP, spanning residues 37 to 203, and generated antibodies against CAP which confirmed the dual localisation (*Figure 1C*). Western blot analysis of parasite lysates revealed the presence of two bands close to the expected size of CAP, which are expressed at a constant level, relative to the *Toxoplasma* loading control, across the first

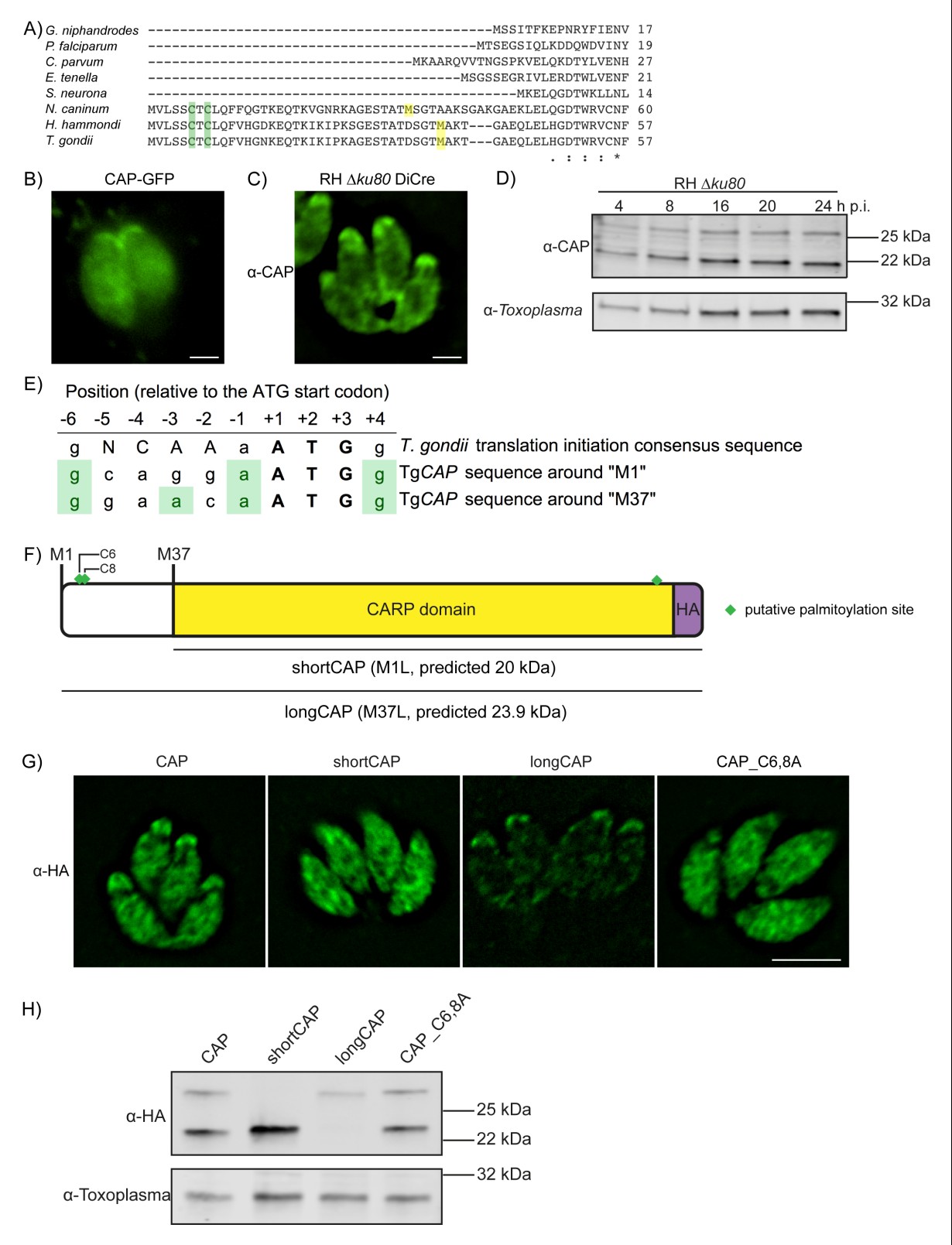

**Figure 1.** Alternative translational start sites lead to the generation of two different CAP isoforms. (**A**) Sequence alignment of the first 57 amino acid residues of *T. gondii* CAP with that of other Apicomplexa. Green shading indicates cysteines which are putative palmitoylation sites, yellow shading indicates methionines which are putative alternative translational start sites. (**B**) Subcellular localisation of a CAP-GFP fusion. Scale bar, 2 μm. (**C**) Subcellular localisation of CAP by immunofluorescence assay (IFA) using rabbit anti-*T. gondii* CAP antibodies. Scale bar, 2 μm. (**D**) Western blot of *CAP*

*Figure 1 continued*

expression levels over the first 24 hr following host cell invasion using anti-*T. gondii* CAP antibodies. Anti-*Toxoplasma* antibodies were used as a loading control. (E) Alignment of the *Toxoplasma* consensus translation initiation (Kozak) sequence (*Seeber, 1997*) with the translation initiation sequences of *CAP*'s first (M1) and second (M37) putative translational start sites. Green shading indicates bases that correspond to the Kozak sequence. (F) Schematic of *T. gondii* CAP with annotations for the two putative CAP isoforms and the mutation experiments performed to test their expression: Mutation of M1 to leucine (L) to produce shortCAP, and mutation of M37 to L producing longCAP. Green diamonds indicate putative palmitoylation sites. (G) IFA and (H) western blot of ectopic HA-tagged CAP isoforms and cysteine mutants (C6 and C8). Inclusion of a HA-tag makes the protein run more slowly than the untagged protein. Anti-*Toxoplasma* antibodies were used as a loading control in (H). Scale bar in (G), 5 μm.

DOI: https://doi.org/10.7554/eLife.50598.002

The following figure supplement is available for figure 1:

**Figure supplement 1.** Endogenous tagging of CAP.

DOI: https://doi.org/10.7554/eLife.50598.003

24 hr following host cell invasion (*Figure 1D*). The dual localisation of CAP could be due to the expression of two isoforms from a single gene, through use of alternate translational start sites, as previously observed for protein kinase G (*Brown et al., 2017*). Indeed, sequence comparison of *T. gondii* CAP to *P. falciparum* CAP reveals a second in-frame methionine at position 37 in *Toxoplasma* that aligns with the *P. falciparum* CAP start methionine (*Figure 1A*). Two additional methionine residues, M71 and M161, are present in the CAP primary sequence. However, these are unlikely used as translational start sites as their products would lead to a truncated and potentially inactive CARP domain.

Translational start sites in eukaryotic mRNAs are preceded by a translation initiation sequence (*Kozak, 1987a*; *Kozak, 1987b*). Consensus translation initiation sequences have been determined for many different organisms and are known as the Kozak sequence (*Nakagawa et al., 2008*). In *Toxoplasma*, the Kozak sequence was elucidated by identifying nucleotides commonly found in the 26 genes assessed (*Seeber, 1997*). This Kozak sequence contains an adenine at the −3 position, relative to the start ATG, which was identified as the most important factor in ribosomal recognition of the start ATG (*Seeber, 1997*). Absence of an adenine can result in ribosome 'leaky scanning' and translation from an internal ATG. We therefore analysed the translation initiation sequence around CAP's first (M1) and second (M37) putative translational start sites. The M1 translation initiation sequence conforms less with the *Toxoplasma* Kozak sequence than the sequence preceding M37; as the former is lacking an adenine at the −3 position (*Figure 1E*). This suggested that alternative translation could lead to the generation of two CAP isoforms: longCAP, which is translated from the first start ATG, and shortCAP, which is translated from the second start ATG. To test this, we generated parasite strains that expressed either the WT sequence or variants where either the first (M1), or the second methionine (M37) of CAP was mutated to leucine, precluding their use as translational start sites (*Figure 1F*). We used the endogenous promotor (i.e. 969 bp upstream of the first start ATG) and introduced the C-terminal HA-tagged CAP variants into the *UPRT* locus of the RH DiCreΔ*ku80*Δ*hxgprt* parasite strain. To determine whether longCAP and shortCAP show differential localisation, as predicted by the presence of putative palmitoylation sites in longCAP, we analysed their subcellular localisation using the HA-tag. While WT CAP parasites showed the expected dual localisation, mutants expressing shortCAP showed exclusively cytoplasmic staining while those expressing longCAP showed predominantly apical staining, with some further signal throughout the parasite (*Figure 1G*). As longCAP contains two putative palmitoylation sites not present in the short-CAP sequence, we next evaluated whether palmitoylation was important for the apical localisation of the long CAP isoform. We mutated the two cysteines in the N-terminus to alanine residues (CAP_C6,8A). CAP_C6,8A appeared cytosolic with no detectable accumulation of CAP at the apical end of the parasites (*Figure 1G*). Western blot analysis of the HA-tagged CAP variants confirmed that WT CAP is identified as two distinct protein bands, which correlate with the predicted size for longCAP and shortCAP (23.9 and 20 kDa). Parasites expressing shortCAP displayed only the lower molecular weight band, while longCAP-expressing parasites only showed the higher molecular weight band. We observed that the protein levels of longCAP appear reduced compared to its isoform in parasites expressing WT CAP, whereas the shortCAP isoform shows an increase (*Figure 1H*). As expected, despite CAP_C6,8A not being detected at the apical end of the parasite, both isoforms were detected by Western blot at levels comparable to parasites expressing WT CAP.

Collectively, these results show that in *Toxoplasma*, and potentially closely related coccidian parasites of *Hammondia* and *Neospora*, CAP is produced as two differentially localised isoforms using alternative translational start sites and the apical localisation of the long isoform is likely palmitoylation dependent.

## Generation of a more stable RH DiCre Δ*ku80* cell line

While attempting to generate a conditional knock out (cKO) of CAP using the DiCre strategy, we observed a frequent loss of one of the DiCre subunits, resulting in dysfunctional floxed CAP parasite strains that lacked the ability to excise CAP. This is possibly because both DiCre subunits are driven by identical 5' and 3' UTRs, allowing for potential recombination in the Δ*ku80* parental line, which possesses an increased efficiency of homologous recombination (*Fox et al., 2009*; *Huynh and Carruthers, 2009*). To prevent loss of DiCre subunits, we generated a new DiCre construct, DiCre_T2A, that expresses the two DiCre subunits from a single promotor using T2A skip peptides (*Kim et al., 2011*). To further minimise the potential for loss of DiCre, we placed a chloramphenicol acetyltransferase (CAT) selectable marker between the two subunits (*Figure 2A*). This would lead to the production of the two separate Cre subunits and the CAT selectable marker. We inserted this construct, into the modified *KU80* locus of the RH Δ*ku80*Δ*hxgprt* strain (*Huynh and Carruthers, 2009*) using CRISPR/Cas9. To test whether expression of the DiCre subunits in the resulting line, RH DiCre_T2A Δ*ku80*Δ*hxgprt*, is stable over time, we integrated the loxP-KillerRed-loxP-YFP reporter construct used in Andenmatten et al., into the *UPRT* locus (*Andenmatten et al., 2013*). As expected, non-treated parasites express KillerRed which, upon RAP treatment, is excised and leads to expression of YFP (*Figure 2B*). As extracellular stress can lead to increased loss of DiCre activity in the original DiCre line (M. Meissner, personal communication, 02.2019), we subjected the new DiCre parasite line (RH DiCre_T2A Δ*ku80*Δ*hxgprt*) to frequent extracellular stress over the course of 65 days. On average, parasites were passaged every 2.3 days, leaving parasites extracellular for ~32 hr in the presence or absence of continuous chloramphenicol selection. We also simultaneously passaged the original DiCre line (*Andenmatten et al., 2013*) under standard, non-stressing, culturing conditions. The new DiCre_T2A line excision efficiency varied between 98% and 99% for replicates on day 1 and was maintained throughout the experiment with a maximal loss of 3% of excision efficiency, irrespective of the presence or absence of chloramphenicol selection, by day 65 (*Figure 2C*). In contrast, the original DiCre line, cultured under standard non-stress conditions, lost 42% of excision capacity by day 65 (*Figure 2—figure supplement 1*), although this was only done as a single replicate. This shows that the second generation DiCre line, RH DiCre_T2A Δ*ku80*Δ*hxgprt*, retains high excision capacity over long periods of time, even when exposed to extracellular stress.

## CAP is important but not essential for in vitro growth and deletion can be largely restored by the short cytoplasmic isoform, but only partially by the membrane bound isoform

To investigate CAP function, we generated a conditional knock out (cKO) of *CAP* using the DiCre strategy. Here, *CAP* with a C-terminal HA tag is flanked by two loxP sites, that recombine upon dimerisation of two split-Cre subunits, a process mediated by the small molecule rapamycin (RAP) (*Andenmatten et al., 2013*). To create the *CAP* conditional knockout line, we integrated a floxed, recodonised and HA-tagged CAP cDNA sequence into the endogenous locus of the RH DiCreΔ*ku80*Δ*hxgprt* line (*Figure 3A*). Correct integration into the locus was confirmed by PCR (*Figure 3B*). However, due to the DiCre issues detailed above, we were not able to successfully induce DiCre-mediated excision of the *CAP* gene. As the DiCre_T2A strategy demonstrated consistently high excision rates over time, during our testing with a reporter construct, we integrated the DiCre_T2A construct into the *KU80* locus of the non-excising floxed *CAP* parasite strain. This generated the parasite line RH DiCre_T2A DiCreΔ*ku80*Δ*hxgprt*_LoxPCAP-HA, called LoxPCAP hereafter. As expected, LoxPCAP displayed dual localisation by IFA (*Figure 3C*). RAP treatment resulted in a complete loss of CAP (ΔCAP), as shown by IFA (*Figure 3C*) and Western blot (*Figure 3E*).

To assess the requirements of CAP and its isoforms for various *Toxoplasma* functions, we complemented LoxPCAP parasites with either wildtype, the short or the long CAP isoform by integration of HA-tagged variants into the *UPRT* locus to generate merodiploid lines (named LoxPCAP$^{CAP}$, LoxPCAP$^{shortCAP}$ and LoxPCAP$^{longCAP}$, respectively). We then excised the endogenous CAP copy by RAP

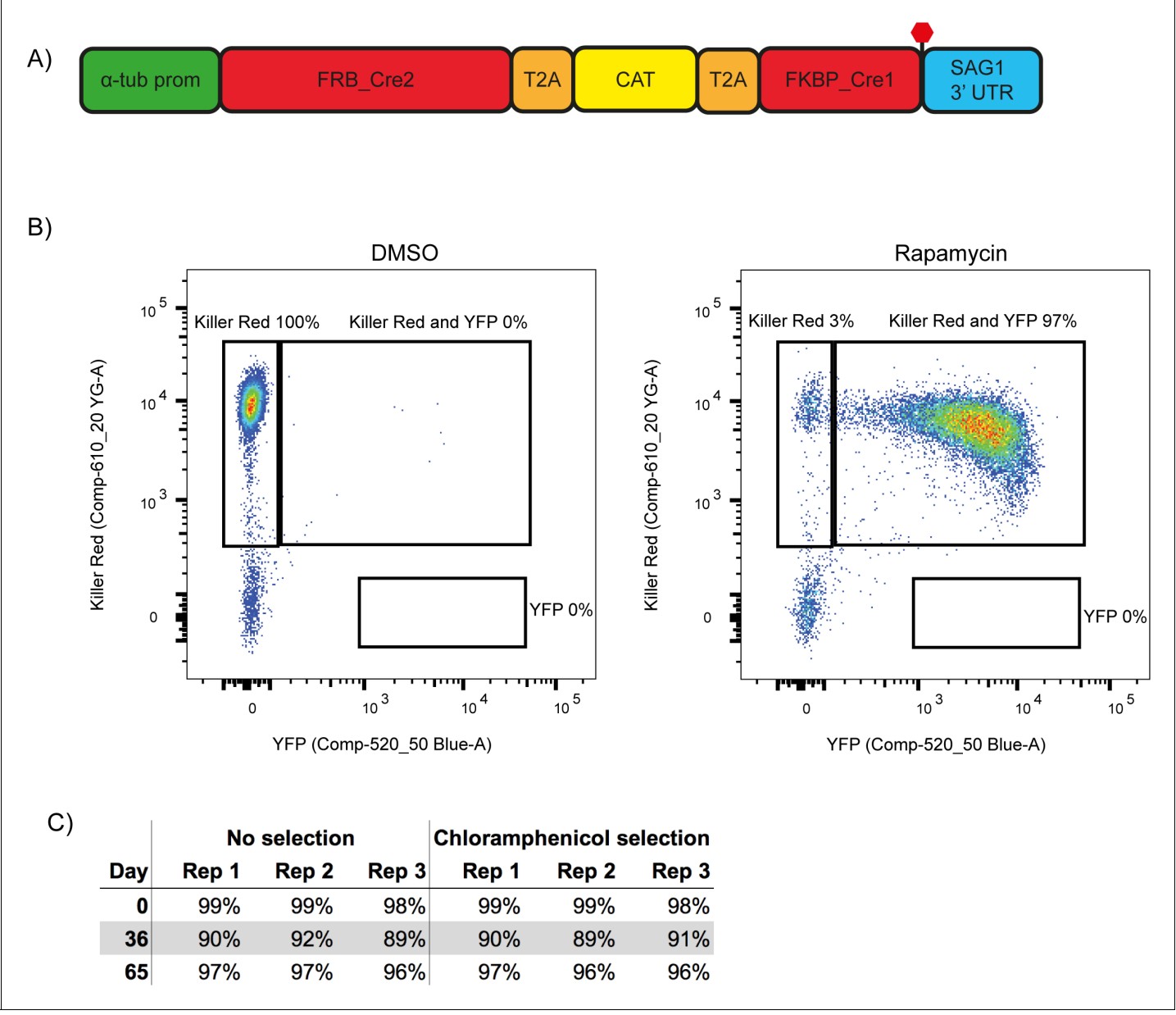

**Figure 2.** A second generation RH Δ*ku80* DiCre_T2A parasite strain stably expresses DiCre. (**A**) Schematic of the DiCre_T2A expression construct. The chloramphenicol resistance cassette (CAT) is flanked by T2A skip peptides. The two Cre subunits (FRB_Cre2 and FKBP_Cre1) are located on either side of the T2A::CAT::T2A cassette. The fusion protein is driven by the alpha-tubulin promotor with a *SAG1* 3' UTR. The red hexagon indicates the position of the stop-codon. (**B**) Flow cytometry analysis to determine excision efficiency of the RH DiCre_T2A Δ*ku80* line following 65 days of frequent extracellular stress. Excision is determined by a shift from Killer Red[(+)] to Killer Red[(+)] and YFP[(+)] expression. Parasites were analysed 22 hr after induction with 50 nM rapamycin (RAP) for 4 hr. Due to analysing 22 hr after induction of excision, parasites still have residual KillerRed signal. (**C**) Table summarising the excision efficiency of RAP treated RH DiCre_T2A Δ*ku80* parasites over time in the presence or absence of chloramphenicol selection. 'Day' refers to the number of days in cell culture while 'rep' corresponds to biological replicates.

DOI: https://doi.org/10.7554/eLife.50598.004

The following figure supplement is available for figure 2:

**Figure supplement 1.** Excision efficiency testing of RH DiCre Δ*ku80*Δ*hxgprt*.
DOI: https://doi.org/10.7554/eLife.50598.005

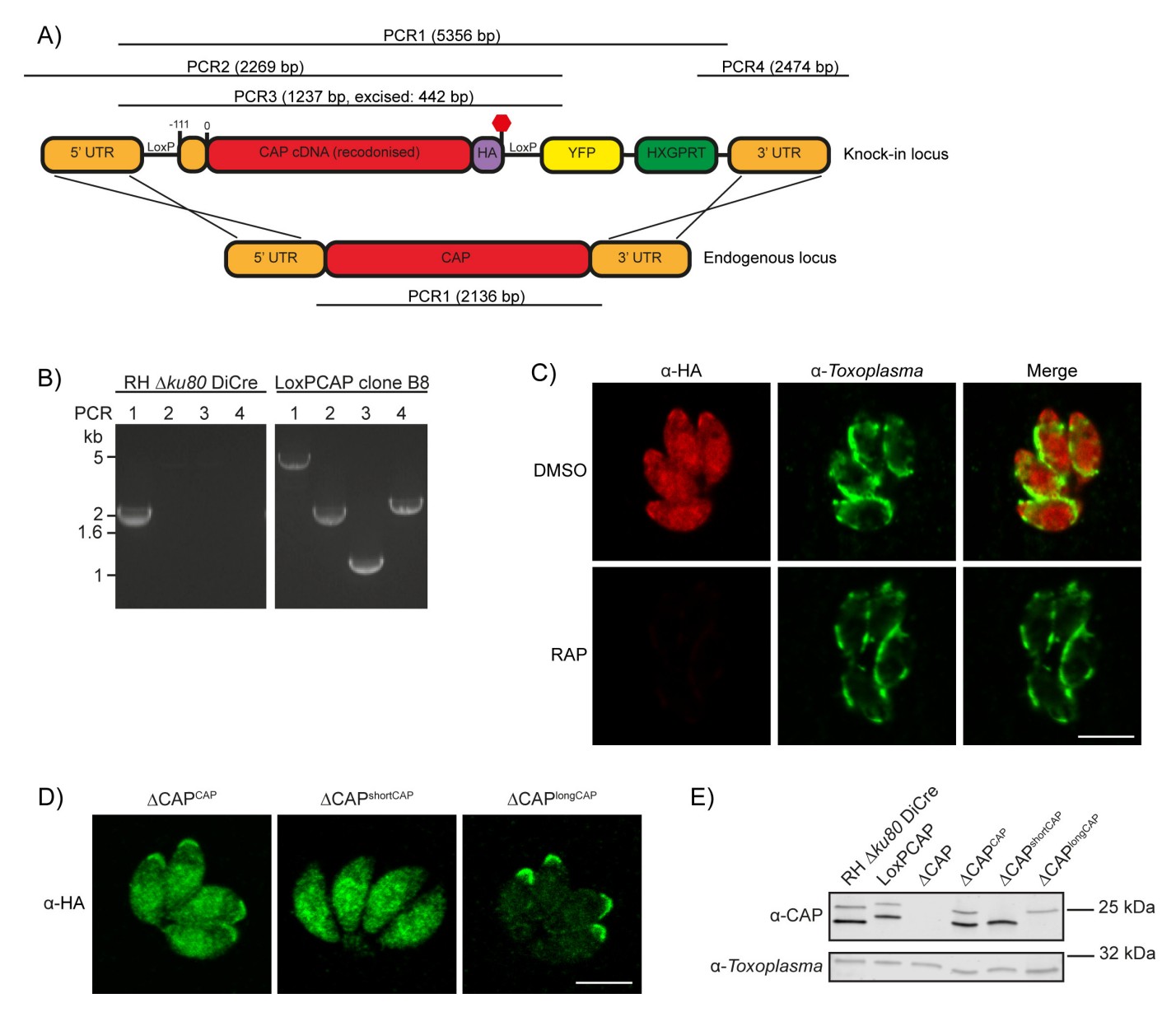

**Figure 3.** Generation of CAP conditional knockout and complementation strains. (**A**) Schematic of the CAP conditional knock out strategy using double homologous integration. The position of the 5' loxP site in the CAP promotor is indicated as well as the predicted sizes of the PCR amplicons. (**B**) Agarose gel showing the expected PCR products for correct integration at the endogenous locus. (**C**) IFA 46 hr after treatment with DMSO or 50 nM rapamycin for 4 hr. Scale bar, 5 μm. (**D**) Subcellular localisation of ectopic HA-tagged CAP isoforms, in the ΔCAP background. Images have been individually contrast adjusted to aid in visualising protein localisation. Scale bar, 5 μm. (**E**) Western blot showing absence of CAP and ectopic HA-tagged CAP isoform expression in the cloned ΔCAP background using anti-*T. gondii* CAP antibodies. Anti-*Toxoplasma* antibodies were used as a loading control.

DOI: https://doi.org/10.7554/eLife.50598.006

The following figure supplements are available for figure 3:

**Figure supplement 1.** Excision testing of the CAP conditional knockout lines.
DOI: https://doi.org/10.7554/eLife.50598.007

**Figure supplement 2.** CAP-HA expression in the ΔCAP^CAP line.
DOI: https://doi.org/10.7554/eLife.50598.008

treatment. Clones were subsequently obtained by limiting dilution and excision verified by PCR (*Figure 3—figure supplement 1*).

This resulted in parasite strains that express only the WT complemented form (ΔCAP$^{CAP}$), the short complemented form (ΔCAP$^{shortCAP}$) or the long complemented form (ΔCAP$^{longCAP}$). We confirmed the differential localisation and translation of these isoforms by IFA (*Figure 3D*) and Western blot (*Figure 3E*). Over the first 24 hr following host cell invasion, no differences in the protein levels of the two CAP isoforms could be observed in the CAP complemented strain (ΔCAP$^{CAP}$), relative to the loading control (*Figure 3—figure supplement 2*). While *CAP* expression levels in the ΔCAP$^{CAP}$ line are comparable to the parental strain (RH ΔKu80 DiCre), longCAP in the ΔCAP$^{longCAP}$ strain appeared to be reduced by ~50%. Although we cannot exclude a functional consequence of the lower long CAP protein levels, as shown further below, long CAP can rescue most phenotypes and reduced levels may therefore not have an overall effect on its function.

Plaquing assays of the ΔCAP parasite line showed no clear reduction of plaque sizes compared to WT parasites (data not shown), however, small differences are not readily observed in these assays. To reliably quantify the contribution of CAP, and its two isoforms independently, to the lytic cycle, we performed a competition assay in which growth of ΔCAP$^{CAP}$ was compared to ΔCAP, ΔCAP$^{shortCAP}$ and ΔCAP$^{longCAP}$. To do so, we integrated an mCherry expressing cassette into the Δku80 locus, replacing the DiCre_T2A cassette in each of these lines. Note, the DiCre_T2A cassette was no longer required because CAP had already been excised in these parasites. After 15 days in growth competition with ΔCAP$^{CAP}$, ΔCAP parasites were largely depleted (>97.3%) from the population, while ΔCAP$^{shortCAP}$ showed a reduction of only 4%. In contrast, ΔCAP$^{longCAP}$ showed an intermediate level of depletion (33.3%). These phenotypes were exaggerated after 30 days in culture; ΔCAP parasites were largely depleted (>99.9%), ΔCAP$^{shortCAP}$ showed a reduction of 12.4%, and ΔCAP$^{longCAP}$ growth was reduced by 43.1% relative to ΔCAP$^{CAP}$ (*Figure 4A*). Collectively, these findings demonstrate that CAP plays an important but non-essential role in cell culture and that its function can be largely restored by the short CAP isoform but only partially by the long isoform.

## CAP contributes to motility, invasion, egress and dense granule trafficking

While the competition assays highlight the importance of CAP during the in vitro lytic cycle, they do not clarify which step of the cycle is affected. To test whether the growth differences between the lines were merely a result of differences in parasite replication rates, we counted the parasites per vacuole for each parasite line. No significant differences were observed (*Figure 4—figure supplement 1*). As CAP is a predicted actin regulator, we next focused our phenotypic analysis on *Toxoplasma* processes for which actin is known to be important, such as motility, egress, invasion and dense granule trafficking (*Heaslip et al., 2016*; *Periz et al., 2017*; *Whitelaw et al., 2017*).

In the absence of CAP (ΔCAP), significantly fewer parasites are able to initiate gliding motility in 3D motility assays (*Leung et al., 2014*) compared to the RH strain (*Figure 4B*). In the motile population, trajectory displacement, trajectory length and maximum achieved speeds were all reduced, while mean speed was not significantly different (*Figure 4B*). To determine which isoform(s) contribute to parasite 3D motility, the motility parameters of ΔCAP$^{CAP}$, ΔCAP$^{shortCAP}$ and ΔCAP$^{longCAP}$ were compared. Both ΔCAP$^{shortCAP}$ and ΔCAP$^{longCAP}$ parasites showed no significant differences in motility initiation, track displacement, track length and speed compared to ΔCAP$^{CAP}$. (*Figure 4—figure supplement 2*). These data indicate that CAP plays a role in initiation of motility and in controlling speed and track length once motile. Complementation with single isoforms shows that initiation of motility can be largely rescued by either CAP isoform and, once motile, either can maintain speed and track length.

As invasion and egress of host cells rely on active motility, we next compared invasion efficiency of ΔCAP and the different complementation lines. ΔCAP showed a reduction of invasion capacity (50.2% reduction compared to the WT complemented line), which was restored by both the short and long isoforms (*Figure 4C*). We also performed egress assays in the presence of 5-Benzyl-3-isopropyl-1H-pyrazolo[4,3-d]pyrimidin-7(6H)-one (BIPPO). BIPPO is a phosphodiesterase inhibitor which results in a strong calcium response in *Toxoplasma* parasites leading to cell motility and synchronised, rapid egress from host cells (*Howard et al., 2015*). ΔCAP parasites showed a substantial delay in egress from host cells at 30 s after induction (73.1% less egress in ΔCAP compared to ΔCAP$^{CAP}$), while at 2 min the majority of parasites have egressed (10.6% less egress in ΔCAP compared to

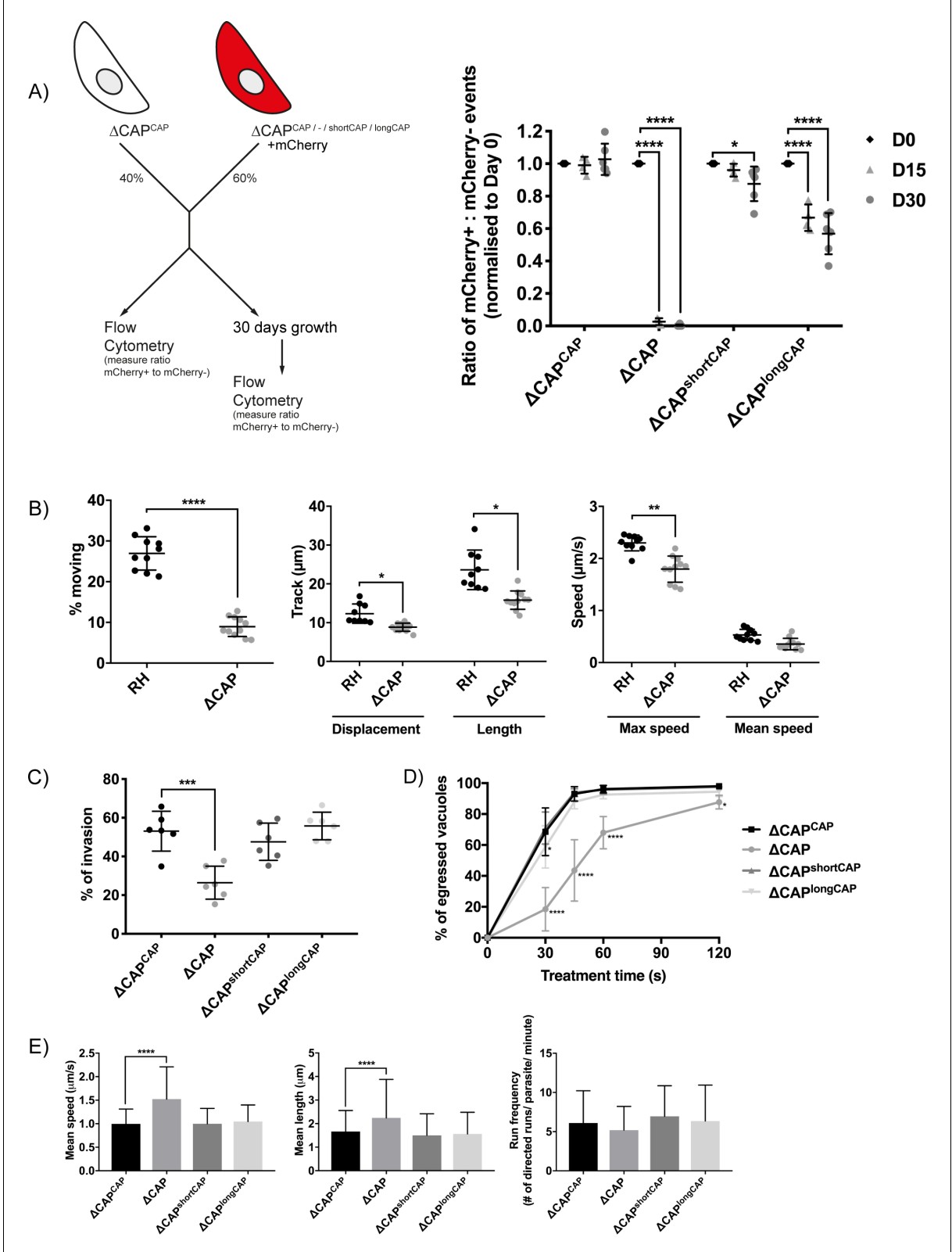

**Figure 4.** CAP plays an important but not essential role during the lytic cycle in cell culture. (A) Overview of the flow cytometry competition assay (left). Competition assays of mCherry-expressing ΔCAP and CAP complementation lines with non-fluorescent WT-complemented parasites (right). The ratio of mCherry[(+)] to mCherry[(-)] parasites was analysed by flow cytometry at day 0, 15 and 30. Data are represented as mean ± s.d. (D0 and D30, $n = 6$. D15, $n = 4$). Two-way ANOVA followed by a multiple comparison Sidak's test was used to compare means between time points. (B) 3D Matrigel-based

*Figure 4 continued on next page*

*Figure 4 continued*

motility assays performed in the absence of inducers of motility. Results are expressed as mean ± s.d. ($n = 4$). Each data point corresponds to a single technical replicate from one of four independent biological replicates, on which significance was assessed using an unpaired $t$-test. (C) Invasion assay comparing ΔCAP to the complemented lines. Data are represented as mean ± s.d. ($n = 3$). One-way ANOVA followed by Dunnett's test was used to compare means to the ΔCAP[CAP] mean (D) Egress assay. Graph shows number of egressed vacuoles in response to BIPPO over time. Data are represented as mean ± s.d. ($n = 3$). Two-way ANOVA followed by Dunnett's test was used to compare means to the ΔCAP[CAP] mean. Stated significance is in comparison to ΔCAP[CAP]. (E) Dense granule trafficking assay. ΔCAP and CAP complemented lines were transiently transfected with SAG1ΔGPI-mCherry that allows visualisation of dense granules. The length and speed of directed runs were recorded using fluorescence microscopy and analysed using ImageJ. Data are represented as mean ± s.d. ($n = 3$). One-way ANOVA followed by Dunnett's test was used to compare means to the ΔCAP[CAP] mean.

DOI: https://doi.org/10.7554/eLife.50598.009

The following source data and figure supplements are available for figure 4:

**Source data 1.** Numerical data of the graphs presented in *Figure 4A,B,C,D,E* and *Figure 4—figure supplements 1* and *2*.

DOI: https://doi.org/10.7554/eLife.50598.012

**Figure supplement 1.** Replication analysis of ΔCAP and the complemented lines.

DOI: https://doi.org/10.7554/eLife.50598.010

**Figure supplement 2.** 3D Matrigel-based motility assays performed in the absence of inducers of motility.

DOI: https://doi.org/10.7554/eLife.50598.011

ΔCAP[CAP]) (*Figure 4D*). This defect was fully restored in ΔCAP[shortCAP] parasites, while ΔCAP[longCAP] showed slightly lower levels of egress after 30 s of treatment but by 60 s were indistinguishable from WT or short CAP complemented lines. We also observed live egress events in which ΔCAP, following host cell egress, showed decreased movement away from the host cell when compared to ΔCAP[CAP], ΔCAP[shortCAP] and ΔCAP[longCAP] (*Videos 1–4*).

A recent study revealed that directed dense granule transport is dependent on filamentous actin (*Heaslip et al., 2016*). Therefore, dense granule trafficking was assessed in ΔCAP and the three complemented lines. We measured the run frequency (# of directed runs/parasite/minute), run length and velocity of directed motions. (*Figure 4E*). Both the mean run length and the mean velocity were significantly increased in ΔCAP (34.8% and 52.9%, respectively), but not in either of the complements. These results indicate that both CAP isoforms play a supporting role in dense granule trafficking and that, upon CAP deletion, dense granules are trafficked further distances at higher

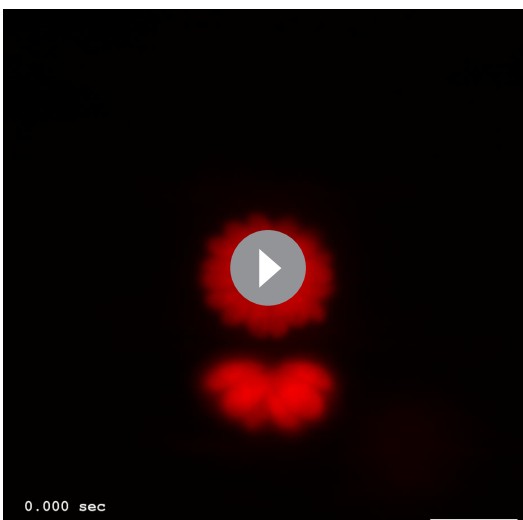

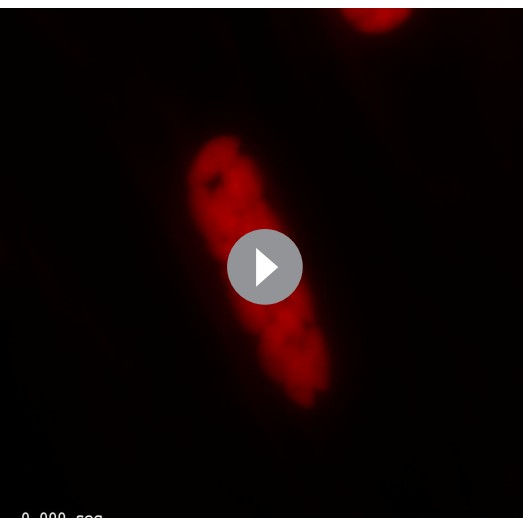

**Video 1.** Live egress imaging of a ΔCAP[CAP] vacuole. Egress was induced by addition of 50 µM BIPPO at 0 s. Image taken every 1.8 s. The video is played at 6.6 fps and the time is indicated in seconds. Scale bar, 10 µm.

DOI: https://doi.org/10.7554/eLife.50598.013

**Video 2.** Live egress imaging of a ΔCAP vacuole. Egress was induced by addition of 50 µM BIPPO at 0 s. Image taken every 1.8 s. The video is played at 6.6 fps and the time is indicated in seconds. Scale bar, 10 µm.

DOI: https://doi.org/10.7554/eLife.50598.014

speed. Collectively these data show that CAP plays a role in the actin-dependent processes described above and that either isoform is able to fulfil these CAP functions.

## ΔCAP parasites display a defect in efficient cell-cell communication

Actin, formin three and myosin I and J have been shown to be important for both parasite rosette organisation and cell-cell communication, as assessed by measuring the rapid transfer of reporter proteins between individual parasites in a vacuole (*Frénal et al., 2017b*; *Periz et al., 2017*; *Tosetti et al., 2019*). Upon CAP deletion, we observed a complete loss of the characteristic rosette formation normally seen in *Toxoplasma*. This aberrant phenotype was fully rescued upon complementation with either WT CAP or the short CAP isoform (*Figure 5A*). ΔCAP^longCAP displayed partial restoration with 44.9% of parasites forming phenotypically normal rosettes while the remainder were disorganised within the vacuole. This was also shown by scanning electron images of infected human fibroblasts in which the host cell and the vacuole membrane was removed ('unroofed') as previously described (*Magno et al., 2005*) (*Figure 5—figure supplement 1*).

It was previously shown that mutant parasite lines which lost rosetting capacity also lost the **r**apid **t**ransfer **o**f **r**eporter **p**roteins (called RTORP hereafter) between them. This has been established by photobleaching of individual parasites in a vacuole, which express a fluorescent reporter protein. Under normal conditions, all parasites in a vacuole contribute to the fluorescence recovery of the photobleached parasite (*Frénal et al., 2017b*; *Periz et al., 2017*; *Foe et al., 2018*; *Tosetti et al., 2019*). Such transfer of reporter proteins has previously been used as a readout for cell-cell communication and, by extension, parasite connectivity (*Frénal et al., 2017b*; *Periz et al., 2017*). To examine whether loss of cell-cell communication also accompanied the disrupted rosetting observed in ΔCAP parasites, we performed fluorescence recovery after photobleaching (FRAP) experiments on ΔCAP and its complementation lines. We chose vacuoles that contained eight parasites/vacuole as at this stage parasites organise in rosettes and individual cells can be easily monitored. We bleached one parasite in the vacuole and recorded both the recovery of fluorescence in the bleached parasite and the fluorescence levels of all other parasites in the vacuole. As expected, the WT-complemented ΔCAP^CAP photobleached parasites display rapid recovery of fluorescence to which, in most cases, all parasites in the vacuole appear to contribute (*Figure 5Bi and C* and *Video 5*). Conversely, ΔCAP photobleached parasites were supported in their rapid fluorescence recovery predominantly by just one other parasite, usually the parasite closest to the bleached cell, resulting in slow recovery (*Figure 5Bii and C* and *Video 6*). In some cases, no recovery was observed in ΔCAP parasites.

Loss of rapid protein transfer between parasites could be explained by a structural disruption of inter-parasite connections provided by the residual body. However, in addition to the residual body, an intravacuolar network (IVN) of tubule-like structures is present in the parasitophorous vacuole. Because of its tubular structure, it could also be involved in cell-cell communication. To determine whether the observed defect in the RTORP was dependent on the presence of the IVN, we performed a FRAP assay in RH Δku80Δgra2 parasites (*Rommereim et al., 2016*) where the IVN fails to form but parasites still organise in rosettes. The results show that RTORP between parasites was not negatively affected (*Figure 5Biii* and *Video 7*). The IVN, therefore, is unlikely involved in cell-cell communication.

To test if the ΔCAP defect in cell-cell communication correlates with an inability to form rosettes, we tested whether the RTORP varied between rosetting (r) and non-rosetting parasites (nr) of the ΔCAP^longCAP line. These experiments revealed that if ΔCAP^longCAP parasites rosette, they show normal RTORP between all cells in the vacuole, while non-rosetting parasites display the same defect as the ΔCAP line, with fluorescence recovery from only one other parasite in the vacuole (*Figure 5C*). Interestingly, Δgra2 vacuoles showed a slight increase in the frequency of photobleach recovery from multiple parasites, but we have not further investigated this phenomenon here. The connectivity of parasites was also assessed based on the percentage of recovery after photobleaching. As expected, ΔCAP and non-rosetting ΔCAP^longCAP photobleached parasites recovered significantly less fluorescence than their rosetting counterparts (*Figure 5D*).

Next, we sequentially photobleached individual ΔCAP parasites in a vacuole and identified which parasites appear to be physically connected based on the transfer of mCherry (*Figure 5E*). This revealed that only parasite pairs in close proximity are rapidly communicating. Collectively the results show that CAP deletion leads to a loss of both rosetting and the rapid transfer of reporter

proteins between more than two parasites. Furthermore, RTORP appears to be dependent on parasite rosette organisation, but not on presence of the IVN.

## Phenotypic characterisation of ΔCAP parasites in daughter cell orientation, synchronised division and apicoplast inheritance

Previously, cell-cell communication has been hypothesised to control synchronous division between parasites within the same vacuole (*Frénal et al., 2017b*). To determine whether ΔCAP parasites display phenotypes previously observed for mutants with defective cell-cell communication, we used IMC3 antibodies to visualise synchronicity of forming daughter cells (*Figure 6A*). At 20 hr post-infection, in the complete absence of rosettes, ΔCAP vacuoles exhibited synchronous division with no significant difference to the WT complemented line (98.3% and 99.6% synchronous division, respectively). (*Figure 6B*). At 24 hr and 30 hr post-infection, synchronous division was reduced to 93.4% and 87.8%, respectively, in the ΔCAP line (*Figure 6B*). IMC3 staining was also used to assess daughter cell orientation. While in WT CAP complemented parasites 94.6% of daughter cells grew in the same orientation, a significant defect was observed in ΔCAP and ΔCAP^longCAP strains; only 42.9% (ΔCAP) and 67.7% (ΔCAP^longCAP) of daughter cells orientated in the same direction. ΔCAP^shortCAP parasites were able to largely overcome this defect, with 89.5% of daughter cells growing in the same orientation (*Figure 6C*). Such improper orientation of daughter cells following budding raises the possibility of improper organelle segregation too. To test this, we looked at apicoplast segregation, another actin-dependent process (*Andenmatten et al., 2013*; *Jacot et al., 2013*). Using streptavidin as a marker for the apicoplast, we identified no significant differences in apicoplast inheritance rates between ΔCAP^CAP and ΔCAP strains (*Figure 6—figure supplement 1*).

## CAP deletion leads to the formation of a decentralised residual body in which all parasites remain connected, despite loss of RTORP

The above data show that despite a loss of rosette organisation and RTORP between tachyzoites in a ΔCAP vacuole, the majority of parasites divide in synchronicity. This was somewhat unexpected as other mutants with defects in rosetting and RTORP display substantially higher defects in synchronous division (*Frénal et al., 2017b*; *Tosetti et al., 2019*). This suggested that ΔCAP parasites still maintain connections that allow flow of proteins or metabolites to synchronise divisions to a much

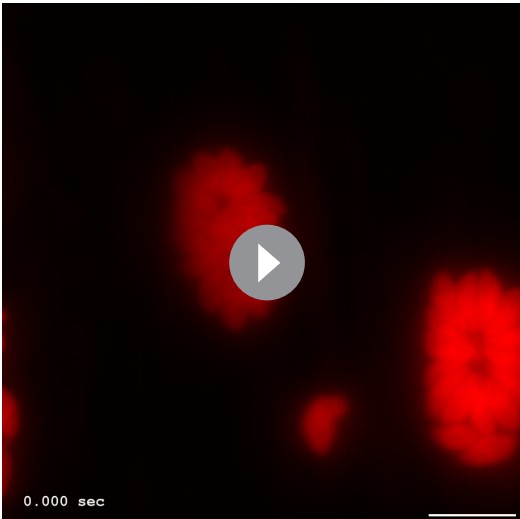

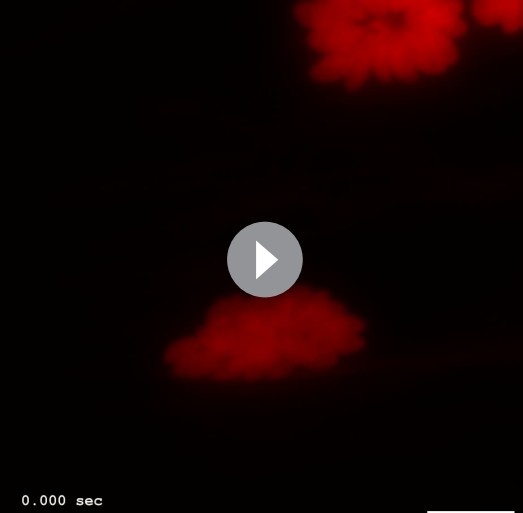

**Video 3.** Live egress imaging of a ΔCAP^shortCAP vacuole. Egress was induced by addition of 50 μM BIPPO at 0 s. Image taken every 1.8 s. The video is played at 6.6 fps and the time is indicated in seconds. Scale bar, 10 μm.
DOI: https://doi.org/10.7554/eLife.50598.015

**Video 4.** Live egress imaging of a ΔCAP^longCAP vacuole. Egress was induced by addition of 50 μM BIPPO at 0 s. Image taken every 1.8 s. The video is played at 6.6 fps and the time is indicated in seconds. Scale bar, 10 μm.
DOI: https://doi.org/10.7554/eLife.50598.016

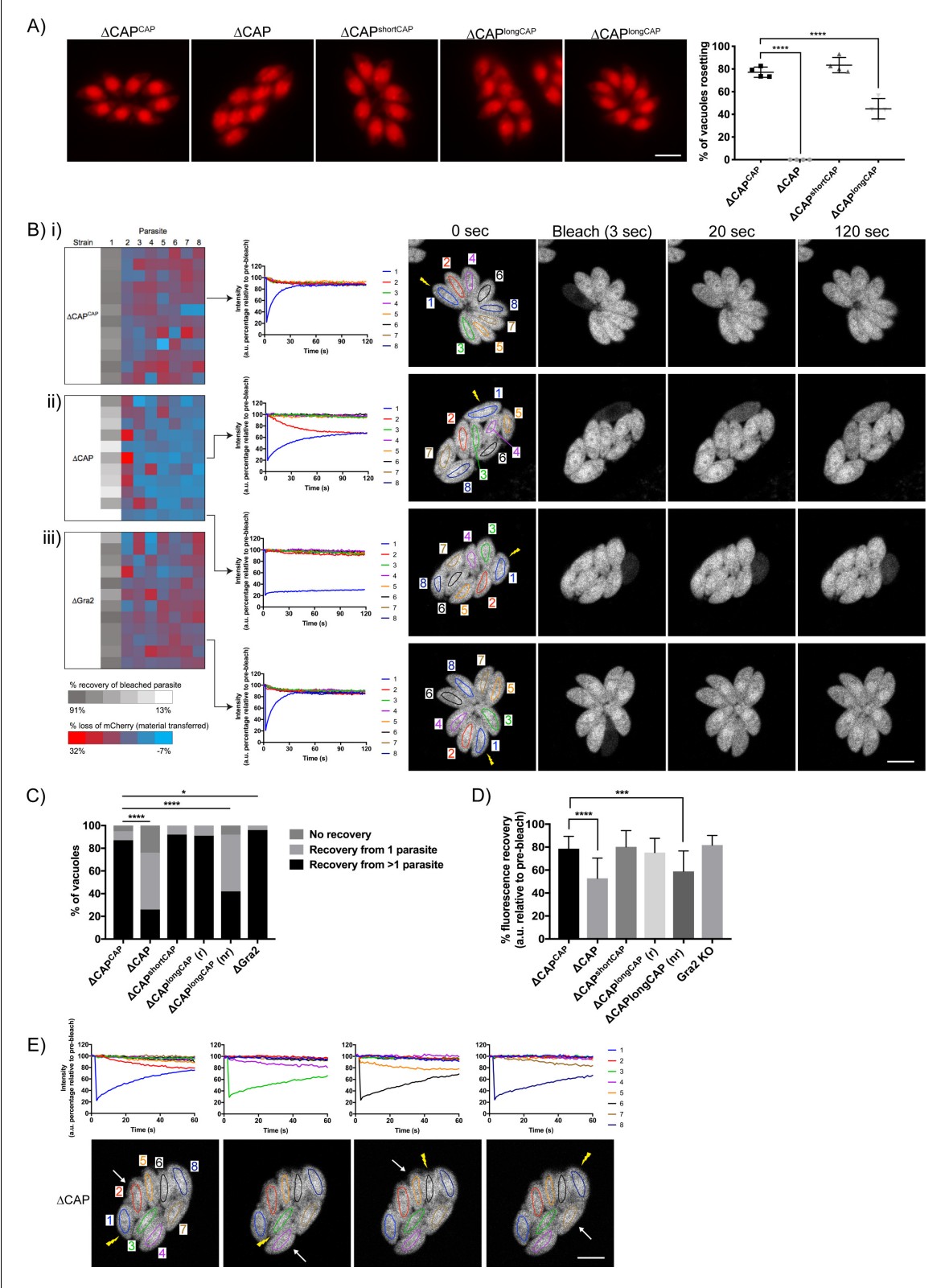

**Figure 5.** CAP is important for rosetting and rapid cell-cell communication. (**A**) Representative fluorescence images of mCherry expressing ΔCAP parasites and complemented isoforms (left). Quantification of rosetting vs. non-rosetting parasites (right). Data are represented as mean ± s.d. (*n* = 2). One-way ANOVA followed by Dunnett's test was used to compare means to the ΔCAP^CAP mean. (**B–E**) FRAP experiments to measure transfer of mCherry between individual parasites in a vacuole. (**B**) Heatmaps showing the percentage change in fluorescence of individual parasites in (**i**) CAP

*Figure 5 continued on next page*

*Figure 5 continued*

complemented, (ii) ΔCAP and (iii) Δgra2 parasite lines. A recovery plot and image for representative vacuoles are included. Regions of interest are numbered; the bleached parasite is '1'. Numbers are allocated based on proximity to the bleached parasite. The yellow lightning bolt indicates which parasite was photobleached. Data are representative of two independent experiments. (C, D) Quantification of the type (C) and amount (D) of recovery for all parasite lines. (ΔCAP and ΔCAP^CAP n = 4, all other lines, n = 2). For (D) data are represented as mean ± s.d. Statistical significance was assessed by either Chi-square test (C) or one-way ANOVA followed by Dunnett's test to compare means to the ΔCAP^CAP mean (D). (E) FRAP analysis of ΔCAP parasites. The images and graphs represent sequential photobleaching and recovery measurements of individual parasites within the vacuole. The yellow lightning bolt indicates which parasite was photobleached. The white arrow identifies the parasite from which the majority of recovery was observed. All scale bars, 5 µm.

DOI: https://doi.org/10.7554/eLife.50598.017

The following source data and figure supplement are available for figure 5:

**Source data 1.** Numerical data of the graphs presented in *Figure 5A,C and D*.
DOI: https://doi.org/10.7554/eLife.50598.019
**Figure supplement 1.** Scanning electron micrographs of intracellular ΔCAP and complemented lines.
DOI: https://doi.org/10.7554/eLife.50598.018

greater level. To investigate this, we established a connectivity map of parasites in a vacuole using correlative light and electron microscopy (CLEM). This allowed us to first analyse connectivity of parasites based on FRAP analysis, and then reconstruct a 3D electron microscopy image of the parasites and their connections by focused ion beam scanning electron microscopy (FIB SEM).

In the ΔCAP^CAP strain, as expected for a WT complemented line, all parasites in the vacuole contribute to recovery of a photobleached parasite (*Figure 7A*) and a normal centralised residual body is formed, connecting all parasites in the vacuole (*Figure 7B* and *Video 8*), with one tubular extension extending away from the residual body with no apparent connections at the distal end. In contrast, for the ΔCAP strain, fluorescence recovery of the photobleached parasite was predominantly observed from those cells in close proximity (*Figure 7C*). To investigate how these parasites are connected, we analysed FIB SEM images obtained from the photobleached vacuole in *Figure 7C*. This revealed membrane bound tubular connections of approximately 300 nm thickness between parasites, despite the aberrant transfer of mCherry and loss of rosetting (*Figure 7Di and ii*). Following the lumen of the connections across three dimensions demonstrates that all parasites in the vacuole are connected by these tubular connections, likely representing a decentralised residual body that forms as a result of an inability to keep the posterior ends of the parasites in close proximity (*Figure 7D.iii*, *Figure 7—figure supplement 1* and *Video 9*). Correlation of the FRAP data with the FIB SEM images showed that rapid transfer of mCherry was always between parasites in close proximity at their basal ends.

While ΔCAP parasites appear to be connected via a decentralised residual body, it could be that transfer of material through these connections is limited by physical barriers, such as the mitochondria which have previously been observed in the residual body (*Frénal et al., 2017b*). Close examination of the residual body connections showed only one such connection contained two mitochondria and a constriction of the decentralised residual body lumen to ~50 nm, while all other connections appeared free of large physical barriers, indicating that this is unlikely an explanation for the lack of RTORP between parasites. However, we observed a complex network of tubules and sheet-like structures, likely representing the endoplasmic reticulum, in the connections (*Figure 7E*) (*Puhka et al., 2012*; *Schroeder et al., 2019*; *Tomavo et al., 2013*; *West et al., 2011*). One of these tubular structures, tracked in three

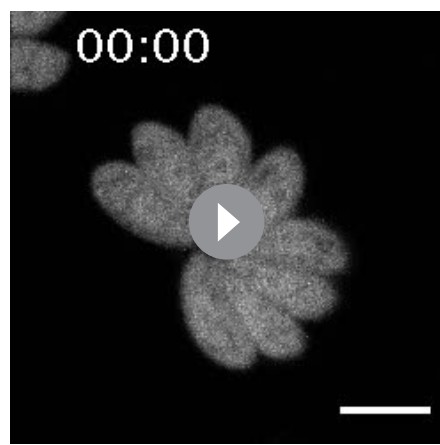

**Video 5.** FRAP on a ΔCAP^CAP vacuole. A single parasite was photobleached at 3 s. Images taken every 1 s. The video is played at two fps and the time is indicated in seconds. Scale bar, 5 µm.
DOI: https://doi.org/10.7554/eLife.50598.020

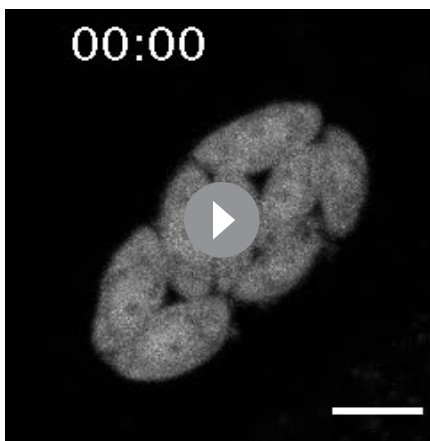

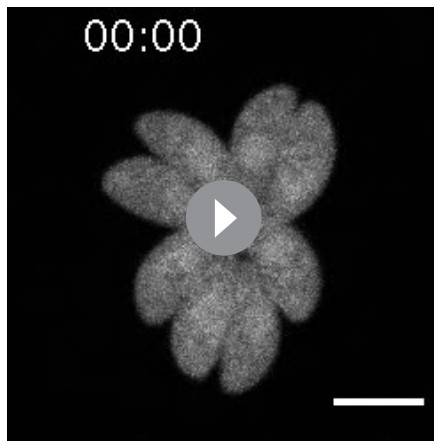

**Video 6.** FRAP on a ΔCAP vacuole. A single parasite was photobleached at 3 s. Images taken every 1 s. The video is played at two fps and the time is indicated in seconds. Scale bar, 5 μm.
DOI: https://doi.org/10.7554/eLife.50598.021

**Video 7.** FRAP on a ΔGra2 vacuole. A single parasite was photobleached at 3 s. Images taken every 1 s. The video is played at two fps and the time is indicated in seconds. Scale bar, 5 μm.
DOI: https://doi.org/10.7554/eLife.50598.022

dimensions, was shown to enter multiple parasites via the basal pole, suggesting a possible continuum between them, which was also observed in the WT CAP complemented parasites. This has not been further examined here but could contribute to exchange of material between parasites.

Collectively, while CAP is important for rosette organisation of parasites, it is not essential for forming and sustaining a residual body with connectivity to all parasites in the vacuole. Furthermore, it shows that the RTORP is not an indicator of parasite connectivity.

## CAP deletion leads to loss of an actin polymerisation centre and extensive actin filamentation in the vacuole

Recently, actin chromobodies (Cb) have been used in *Toxoplasma* to visualise F-actin in both the parasites and the vacuole, with considerable accumulations seen within the residual body (*Periz et al., 2017*) (*Figure 8A*). Given CAP's hypothesised actin regulatory function, as well as its importance in forming a centralised residual body, we sought to determine whether CAP is playing a role in regulating these F-actin structures. For live visualisation of F-actin, we utilised a line expressing an emerald fluorescent protein fused to F-actin chromobodies (Cb-EmFP) (*Periz et al., 2017*). CAP was deleted in the Cb-EmFP line through gene replacement with an mCherry expressing construct, creating the strain Cb-EmFP ΔCAP (*Figure 8B*). Next, we complemented a Cb-EmFP ΔCAP clone with either wildtype, the short or the long CAP isoform by integration of HA-tagged variants into the *UPRT* locus to generate Cb-EmFP ΔCAP^CAP, Cb-EmFP ΔCAP^shortCAP and Cb-EmFP ΔCAP^longCAP, respectively. Loss of CAP expression and successful complementation was confirmed by WB (*Figure 8C*). In agreement with our previous findings, deletion of CAP in the actin chromobody-labelled cells led to complete loss of parasite vacuole rosetting. This phenotype was rescued by complementation with WT CAP or the short CAP isoform, while long CAP isoform complementation resulted in 42% of vacuoles rosetting (*Figure 8D and E*). After three rounds of division (eight parasites/vacuole) and onwards, loss of rosetting was accompanied by extensive filamentous structures of the chromobody throughout the parasitophorous vacuole (*Figure 8D*).

Previous studies have noted an actin polymerisation centre in the apical juxtanuclear region of the parasite (*Periz et al., 2017*; *Tosetti et al., 2019*). We observed a complete loss of the juxtanuclear actin polymerisation centre in the Cb-EmFP ΔCAP line, suggesting CAP plays a role in establishing this intracellular F-actin accumulation (*Figure 8D*). WT CAP and shortCAP complementation restored this actin polymerisation centre. LongCAP complementation only rescued it in 11% of parasites, irrespective of rosetting, which shows that the juxtanuclear actin accumulation and rosetting are not phenotypically linked (*Figure 8E*).

It has previously been shown that BIPPO stimulation of extracellular parasites leads to F-actin accumulation at the basal end of parasites (*Periz et al., 2017*; *Tosetti et al., 2019*). BIPPO

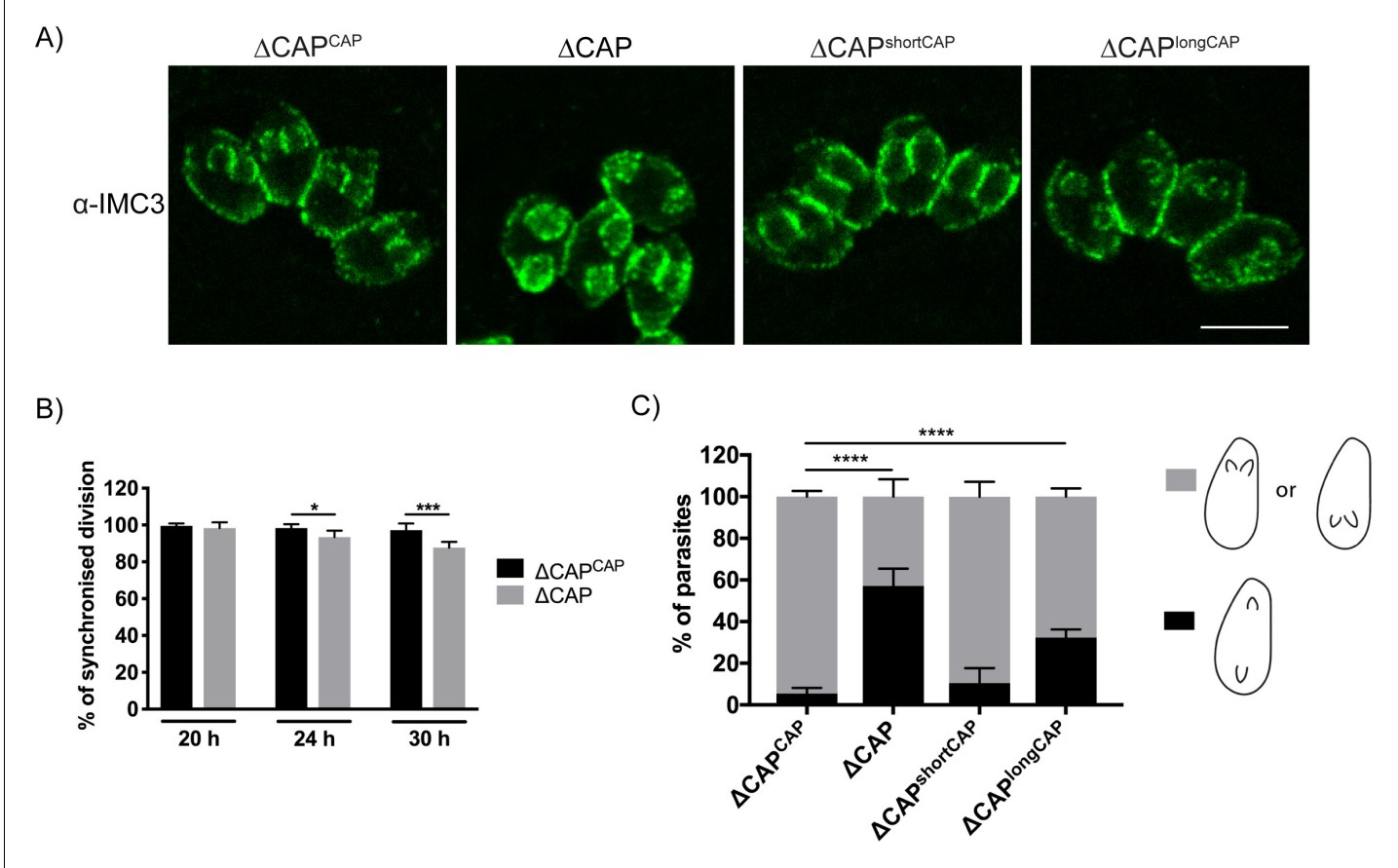

**Figure 6.** CAP is important for daughter cell orientation, but not synchronous division. (A) Parasites stained with anti-IMC three antibodies to visualise daughter cell orientation and division. Scale bar, 5 µm. (B) Quantification of synchronicity of division in parasite vacuoles using daughter cell staining from (A). Data are represented as mean ± s.d. (n = 3). Significance was assessed using an unpaired two-tailed *t*-test. (C) Quantification of daughter cell orientation in parasite vacuoles reveals a defect in the daughter cell orientation of ΔCAP parasites. Data are represented as mean ± s.d. (n = 3). One-way ANOVA followed by Dunnett's test was used to compare means to the ΔCAP^CAP mean.

DOI: https://doi.org/10.7554/eLife.50598.023

The following source data and figure supplement are available for figure 6:

**Source data 1.** Numerical data of the graphs presented in *Figure 6B,C* and *Figure 6—figure supplement 1*.
DOI: https://doi.org/10.7554/eLife.50598.025
**Figure supplement 1.** Apicoplast segregation analysis.
DOI: https://doi.org/10.7554/eLife.50598.024

treatment of both CAP knockout and WT CAP complemented lines led to a basal accumulation of F-actin (*Figure 8F*), suggesting that CAP is not essential for this process.

## F-actin filaments appear to be present in most tubular connections between ΔCAP parasites

The strong accumulation of chromobody positive structures in the vacuole of ΔCAP parasites suggested that F-actin filaments may be located in the decentralised residual body. To visualise F-actin and potentially correlate this to the tubular residual body connections between parasites, we performed CLEM experiments with the chromobody expressing ΔCAP line using serial block face scanning electron microscopy (SBF SEM). In the vacuole we identified regions with a strong chromobody signal and others with much weaker, or no detectable, chromobody signals (indicated by magenta arrowheads) (*Figure 9Ai*). Using the EM images, we followed the tubular connections in the vacuole and overlaid the chromobody signal (*Figure 9Aii and iii*). This analysis showed that the accumulation of chromobody in the vacuole is found within regions of the tubular connections between parasites.

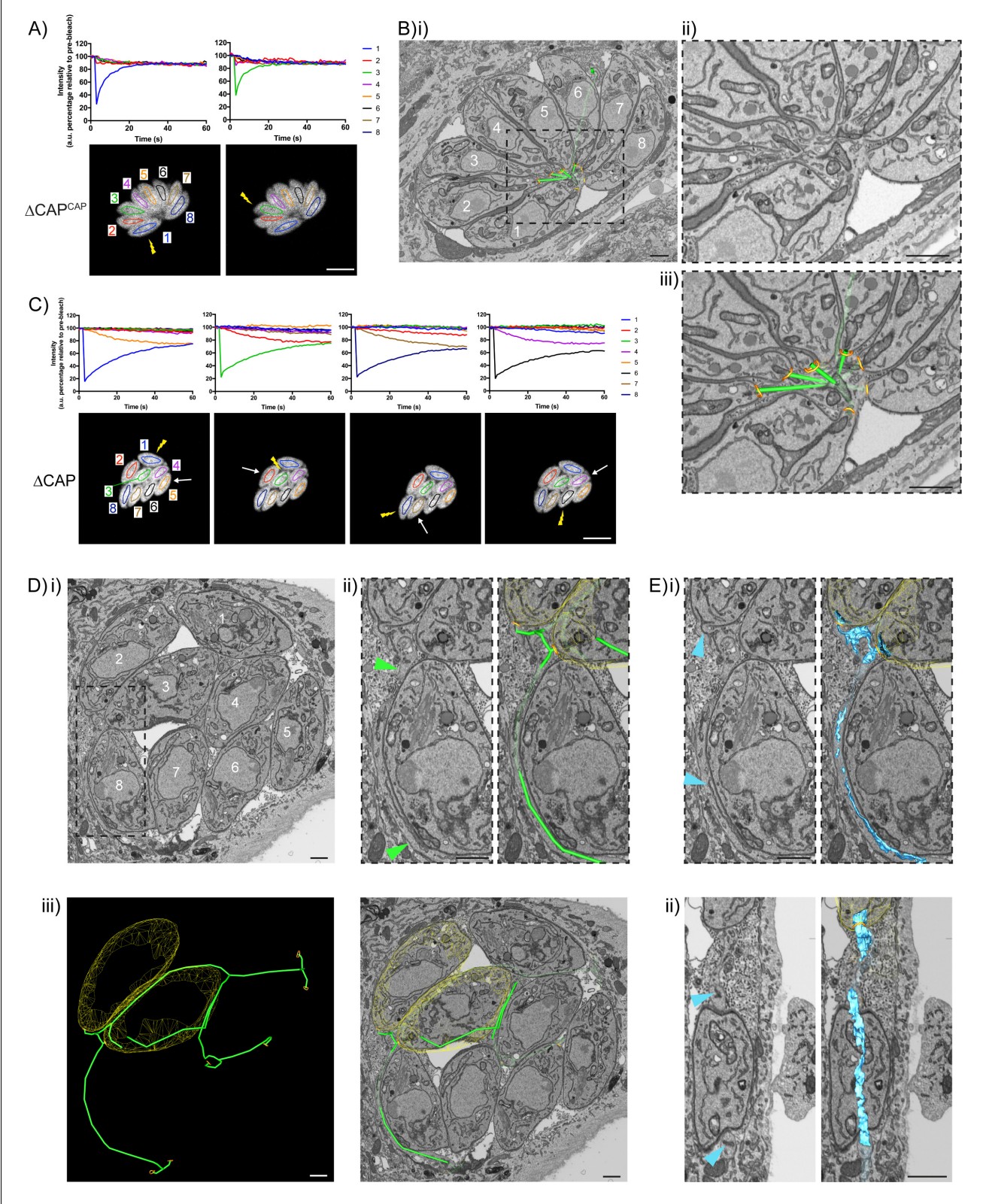

**Figure 7.** CAP KO parasites are still connected by a decentralised residual body. (**A**) FRAP analysis of ΔCAP^CAP parasites. The images and graphs represent sequential photobleaching and recovery measurements of individual parasites within the vacuole. (**B**) (**i**) FIB SEM of the vacuole from (**A**) with a 3D model highlighting the residual body (green skeleton representing how the approximate centre can be traced through to the basal poles) and parasite openings at the basal pole (orange ring). Note that the illustrated connections display the distance between the parasite's posterior ends, and

*Figure 7 continued on next page*

*Figure 7 continued*

do not represent tubular connections between them. Numbering of parasites is consistent with (**A**). (**ii**) Zoomed image of the residual body from (**i**), (**iii**) zoomed image of the residual body from (**i**) overlaid with the 3D model. (**C**) FRAP analysis of ΔCAP parasites. The images and graphs represent sequential photobleaching and recovery measurements of individual parasites within the vacuole. (**D**) (**i**) FIB SEM of the vacuole from (**C**), numbering of parasites is consistent with (**C**). (**ii**) Zoomed image of the boxed region from (**i**). Green arrows indicate the residual body (left panel). A 3D model highlights the residual body (green skeleton drawn through the lumen of the connections between parasites), parasite openings at the basal pole (orange ring) (right panel). (**iii**) A 3D model of selected features in the vacuole from (**C**), highlighting the residual body (green), parasite openings at the basal pole (orange ring) and coarse segmentation of two of the parasites (yellow) (left panel). This model is shown with an orthoslice from the FIB SEM volume (right panel). (**E**) Zoomed images of the boxed region from (**Di**). Blue arrows indicate a putative ER structure (left). A 3D model highlights this putative ER (blue) and parasite openings at the basal pole (orange ring) (right). (**i**) View facing a Z orthoslice. (**ii**) View facing an X orthoslice. The yellow lightning bolt indicates which parasite was photobleached and the white arrow identifies the parasite from which recovery was observed (**A,C**). Scale bar, 5 μm (**A, C**) or 1 μm (**B, D, E**).

DOI: https://doi.org/10.7554/eLife.50598.026

The following figure supplement is available for figure 7:

**Figure supplement 1.** Close up FIB SEM images of parasite connections in the ΔCAP vacuole from *Figure 7C–E*.

DOI: https://doi.org/10.7554/eLife.50598.027

Interestingly, tubular connections with strong chromobody accumulation appeared morphologically distinct from the other parts of the tubular connections with lower or undetectable chromobody signal. Close examination of tubules with low chromobody signal revealed that they contain membranous structures within the tubule lumen (magenta 3D model and arrows) (*Figure 9Biii and iv*). These membranous structures are reminiscent of those seen in *Figure 7E* for ΔCAP parasites not expressing the chromobody. In contrast, tubule regions with high chromobody signal were essentially devoid of any internal membranous structures (green 3D model and arrows) (*Figure 9Bi and ii*). While morphologically distinct, the connections could nevertheless be traced between the different regions and to the parasite basal pores (*Video 10*). Collectively these results show that the chromobody, and by extension F-actin, is found in most if not all tubular connections, although of varying intensity. It is worth noting that in this particular vacuole, all parasites appear to be synchronised in their division and the daughter cells are not uniformly orientated. This largely resembles the phenotype observed for the non-chromobody ΔCAP strain.

## Deletion of CAP results in completely avirulent type II parasites, but not in the type I RH strain

The short CAP isoform complements most phenotypes in cell culture while the long CAP isoform, in most cases, only shows a partial rescue. This raises the question about the evolutionary roles of the two different isoforms. To better discriminate the functions of the short and longCAP isoforms, we wanted to examine their respective roles in natural infections, where parasites encounter a number of additional stresses, including shear stress and the immune system. Accordingly, we addressed the essentiality of CAP and its isoforms in mouse infections. We hypothesised that if both isoforms are essential for parasite survival in a natural host, loss of either of the two isoforms would manifest in a fitness cost. First, we injected male C57BL/6 mice with ~25 ΔCAP or ΔCAP^CAP tachyzoites of the virulent RH strain, and monitored them over the course of 12 days. In both instances mice

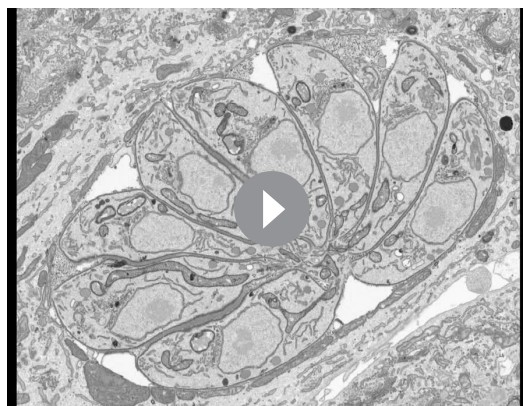

**Video 8.** FIB SEM of the ΔCAP^CAP vacuole from *Figure 7A–B* with 3D model overlay. All FIB SEM slices of the ΔCAP^CAP vacuole *Figure 7A–B*. The 3D model highlights the residual body (green skeleton representing how the approximate centre can be traced through to the basal poles, along with a structure that extends from that centre but has no connection) and parasite openings at the basal pole (orange ring). The volume of the FIB SEM dataset shown is indicated in the Materials and methods.

DOI: https://doi.org/10.7554/eLife.50598.028

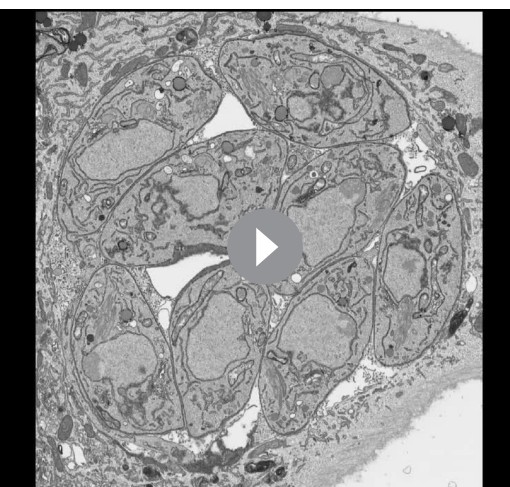

**Video 9.** FIB SEM of the ΔCAP vacuole from *Figure 7C–E* with 3D model overlays. All FIB SEM slices of the ΔCAP vacuole *Figure 7C–E*. The 3D model highlights the residual body (green skeleton drawn through the approximate central axis of the lumen of the connections between parasites), parasite openings at the basal pole (orange ring), coarse segmentation of the central part of two of the parasites (yellow), and part of the putative ER (blue, note that only a selected region was segmented; from the inner face of two of the basal pores through part of the decentralised residual body). The volume of the FIB SEM dataset shown is indicated in the Materials and methods.
DOI: https://doi.org/10.7554/eLife.50598.029

began to succumb to infection after 9 days, indicating that, despite the motility and rosetting phenotypes, CAP depletion in a type I RH background still results in high virulence in mice (*Figure 10A*). As RH parasites have frequently been associated with hypervirulence and do not form cysts in vivo, we next generated ΔCAP parasites and complemented versions in the type II Pru strain. In stark contrast to the RH line, upon injection of ~50,000 tachyzoites, Pru ΔCAP parasites showed no virulence in mice while the Pru ΔCAP$^{CAP}$, Pru ΔCAP$^{shortCAP}$, and Pru ΔCAP$^{longCAP}$ complemented parasites led to a lethal infection, with most mice succumbing to the parasites 8–10 days post-infection (*Figure 10B*). These data show that expression of either individual isoform is sufficient to cause a lethal infection. Next, to look at formation of tissue cysts, which leads to a chronic infection, we injected a lower dose of ~5000 tachyzoites of the Pru lines into mice. The majority of mice survived until day 32 post-infection, although 2 ΔCAP$^{CAP}$-infected mice and 1 ΔCAP$^{shortCAP}$-infected mouse died before reaching this endpoint. At day 32, the mice were sacrificed, brain samples were collected and serum tested for anti-*Toxoplasma* antibodies, confirming that all mice were successfully infected with *Toxoplasma* (*Figure 10C*). Both the Pru ΔCAP and Pru ΔCAP$^{longCAP}$ infections demonstrated considerably lower cyst loads compared to Pru ΔCAP$^{CAP}$ and Pru ΔCAP$^{shortCAP}$ infected mice (*Figure 10D*). These results show that CAP plays an essential role in the virulence of the type II Pru parasite strain, but not the type I RH strain. Moreover, while the short CAP isoform is able to fulfil all functions of CAP in cell culture and in the mouse model, the long isoform, despite its ability to complement most phenotypes to at least ~50% of WT complement levels, has a significant defect in establishing a chronic infection at the infectious dose used here.

## Discussion

*Toxoplasma* actin is important for a range of cellular processes, from organelle segregation and cell-cell communication, to gliding motility: a crucial factor in parasite dissemination (*Andenmatten et al., 2013*; *Egarter et al., 2014*; *Periz et al., 2017*). Despite the key role of actin in parasite biology, our understanding of actin dynamics and its regulation remains incomplete. CAP is a ubiquitous protein with a conserved role in regulating actin dynamics. In this study, we have established that *Toxoplasma* CAP, while dispensable for growth in cell culture, plays an important role in a range of actin-dependent processes. CAP is expressed by an alternative translation initiation site, giving rise to two independent isoforms: shortCAP and longCAP. Through sequence alignment we identified that the alternate translation initiation site is conserved only within *Toxoplasma*, *Neospora and Hammondia*. These are all members of the Toxoplasmatinae, a subfamily of the Apicomplexa phylum, suggesting that while CAP is present in all apicomplexa, the long isoform is specific to *Toxoplasma* and its closest relatives.

Here we investigated the role of CAP and both CAP isoforms in the *Toxoplasma* lytic cycle. Through in vitro competition assays, we show that complementation with shortCAP is enough to overcome the majority (87.6%), but not all, of the growth defect associated with CAP depletion. This suggests that longCAP is performing a specific function important for parasite fitness, which

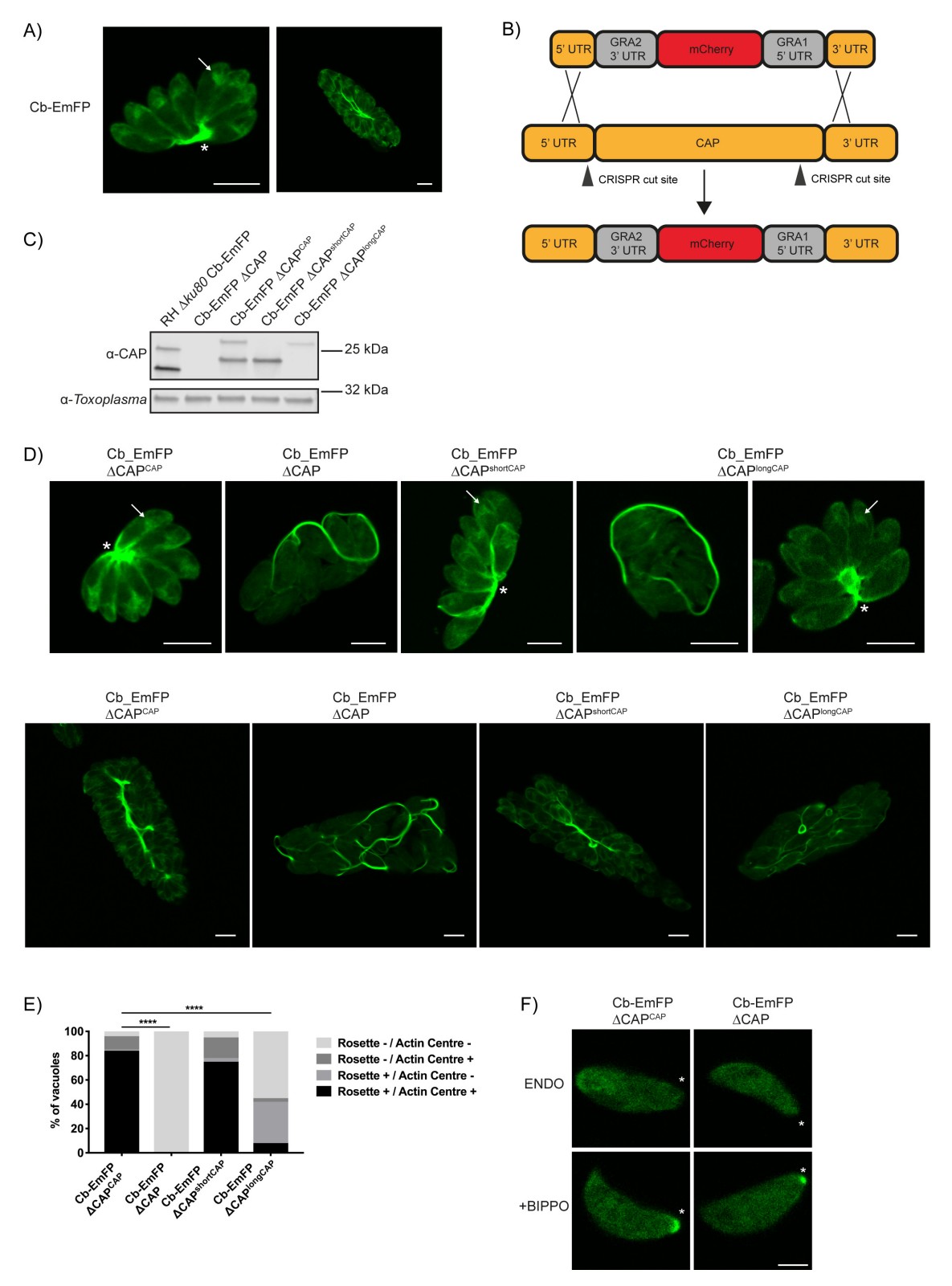

**Figure 8.** CAP deletion impacts some but not all F-actin structures in *Toxoplasma*. (**A**) Live microscopy of the RH *Δku80* Cb-Emerald parental parasite strain. Arrows indicate an actin polymerisation centre. Asterisks indicate strong Cb-Emerald signal at the residual body. Scale bar, 5 μm. (**B**) Schematic of the CAP knockout strategy in the Cb-Emerald expressing strain. Arrowheads indicate CRISPR/CAS9 cut sites. (**C**) Western blot of *CAP* expression levels using anti-*T. gondii* CAP antibodies. Inclusion of a HA-tag makes the protein run more slowly than the untagged protein. Anti-*Toxoplasma*

*Figure 8 continued on next page*

*Figure 8 continued*

antibodies were used as a loading control. (D) Live microscopy of Cb-Emerald parasites with CAP modifications. Upper panels show vacuoles containing eight parasites. For longCAP expressing parasites a rosetting and one non-rosetting vacuole is shown. Lower panels show vacuoles with >32 parasite. Arrows indicate an actin polymerisation centre. Asterisks indicate strong Cb-Emerald signal at the residual body. Arrowheads highlight the extensive actin structures in the vacuole. Scale bar, 5 μm. (E) Quantification of vacuole rosetting and apical juxtanuclear actin polymerisation centre frequency. Statistical significance was assessed by Chi-square test ($n = 2$). (F) Cb-Emerald accumulation at the basal end of BIPPO-stimulated parasites can still be observed in ΔCAP parasites. Asterisks indicate the basal end of the cell. Note: each panel is a different parasite. Scale bar, 2 μm.

DOI: https://doi.org/10.7554/eLife.50598.030

The following source data is available for figure 8:

**Source data 1.** Numerical data of the graph presented in *Figure 8E*.

DOI: https://doi.org/10.7554/eLife.50598.031

---

shortCAP is unable to compensate for. Furthermore, longCAP complementation restored 56.9% of the growth defect, arguing for a degree of functional overlap between the two isoforms. Our experiments have not uncovered a phenotype that only longCAP can rescue, which would have revealed a unique function. Its apical localisation, however, makes it tempting to speculate a function in actin regulation during invasion, which is a phenotype it can fully rescue. It could be that higher actin turnover is required at the apex. The concentration of longCAP at the apex could fulfil this need by influencing G-actin levels required for Formin one dependent actin dynamics (*Tosetti et al., 2019*). Indeed, it has previously been hypothesised in other organisms that increased local concentration of CAP results in the sequestration of actin monomers (*Ono, 2013*). However, providing evidence for this hypothesis is limited by the fact that the shortCAP isoform can rescue all phenotypes in cell culture under the conditions tested. Given that cell culture assays do not fully represent the environment *Toxoplasma* normally encounters, we aimed to tease apart the functions of the two isoforms in mice. Surprisingly, even here the short isoform appears able to compensate for the lack of longCAP. This suggests that even under conditions encountered in the natural host, the longCAP isoform plays only a minor role, although dosage effects or routes of infection may well be confounding factors when assessing virulence of the different strains. We cannot rule out that the differences in proteins levels of longCAP and shortCAP in the single isoform producing lines are affecting the phenotypic observations. However, both isoforms are able to rescue most phenotypes despite longCAP being expressed at lower levels than the parental strain, indicating that the protein levels do not appear to substantially affect CAP function.

Through use of the actin chromobody-expressing line, we show a link between CAP and F-actin. CAP deletion not only results in loss of an actin polymerisation centre but also the appearance of extensive actin filaments within the parasitophorous vacuole. Furthermore, CAP depletion led to loss of the juxtanuclear actin polymerisation centre but, interestingly, had no impact on apicoplast segregation which occurs in the same region. This is somewhat of an unexpected result given the actin-dependent nature of apicoplast segregation, along with the previous published results that both Formin two and ADF mutants display a loss of the juxtanuclear actin centre and defects in apicoplast segregation (*Jacot et al., 2013*; *Rosario et al., 2019*; *Stortz et al., 2019*; *Tosetti et al., 2019*). An explanation for efficient apicoplast inheritance in the ΔCAP line, despite loss of the visible actin polymerisation centre, could be that F-actin dynamics are not completely abrogated. It is likely that Formin 2-mediated actin nucleation is still occurring in this region and is sufficient to ensure apicoplast segregation. Similar to Formin two depletion, Cb-EmFP ΔCAP parasites show loss of the juxtanuclear actin polymerisation centre and abnormal daughter cell orientation. In summary, this data suggests that the juxtanuclear actin polymerisation centre is not required for apicoplast inheritance, but may be required for the positioning of daughter cells during their formation.

One of the most pronounced phenotypes of CAP deletion is loss of rosetting: the highly symmetrical physical distribution of parasites within the vacuole. Previous studies on actin, myosin I, myosin J, ADF and Formin three have suggested that the actomyosin motor is important for rosetting (*Frénal et al., 2017a*; *Haase et al., 2015*; *Periz et al., 2017*; *Tosetti et al., 2019*). Our ΔCAP data further supports this hypothesis. Interestingly, complementing ΔCAP parasites with longCAP restored rosetting in just under half of all vacuoles, with the remainder appearing as equally disordered as the ΔCAP vacuoles. Despite this mixed population, we did not observe vacuoles with a combination of organised and disorganised parasites, suggesting rosetting is a binary outcome;

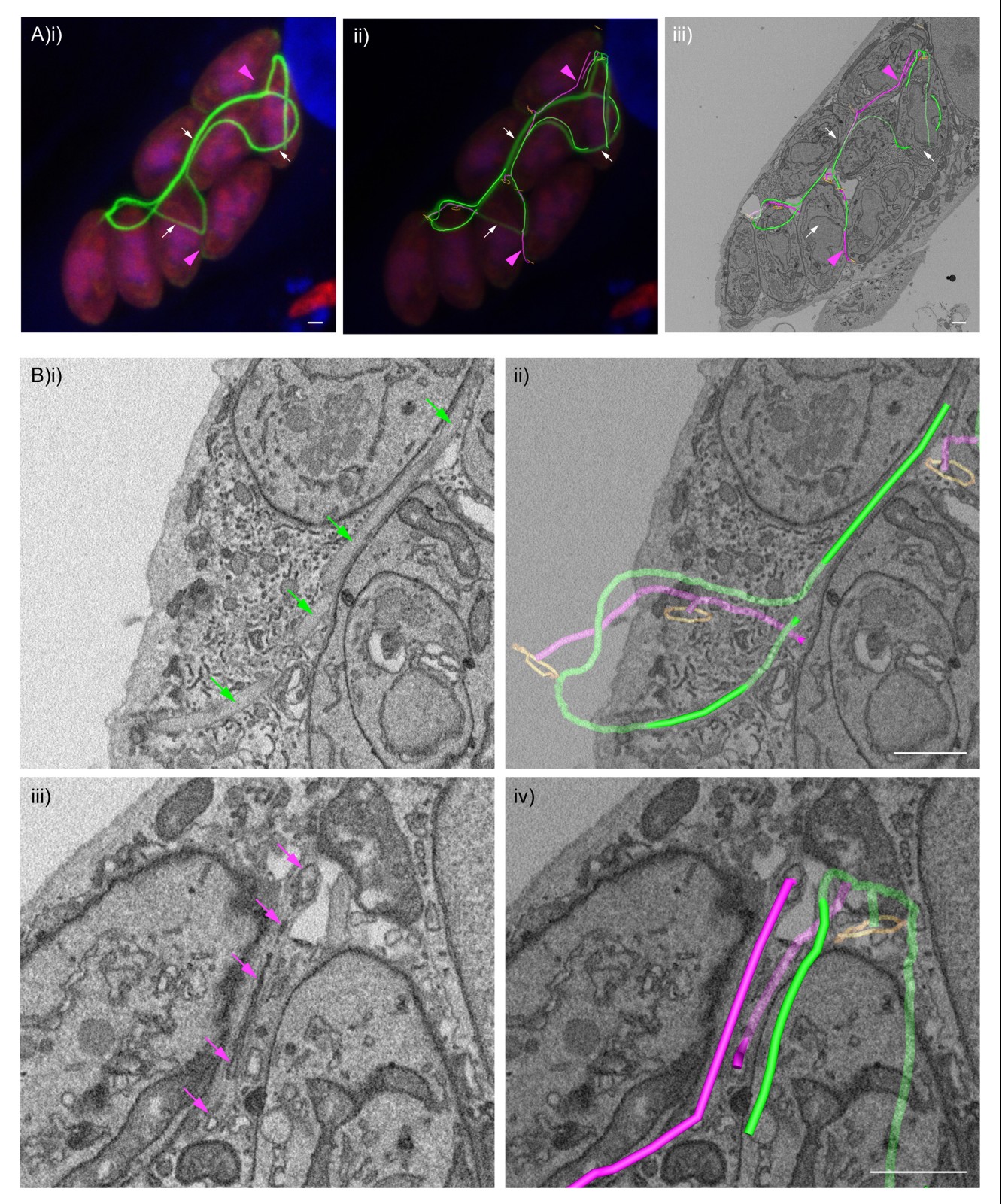

**Figure 9.** F-actin structures localise within the tubular connections of the decentralised residual body. Correlative confocal fluorescence and SBF SEM analysis of Cb-Emerald ΔCAP parasites. (**A**) (**i**) ClearVolume orthogonal XY render of confocal fluorescence microscopy. (**iii**) SBF SEM XY orthoslice of the vacuole from (**i**), with segmented skeleton 3D models highlighting the residual body (tubes of green and magenta). The parasite basal pores are highlighted by orange rings. (**ii**) Coarse overlay of (**i**) and (**iii**). (**A**) White arrows; regions of the Cb-Emerald fluorescence not captured in the SBF SEM

*Figure 9 continued on next page*

*Figure 9 continued*

acquisition. Green 3D model tubes; regions of the residual body with strong Cb-Emerald fluorescence in the corresponding 3D regions of the LM dataset and no internal membranous structures. Magenta arrowheads; regions of the residual body model with corresponding low or undetectable Cb-Emerald fluorescence. Magenta 3D model tubes; regions of the residual body model with corresponding low or undetectable Cb-Emerald fluorescence and internal membranous structures. (**B**) Magnified regions of XY slices 68 (**i and ii**) and 58 (**iii and iv**) from the SBF SEM dataset in (**Aiii**). Morphological differences are indicated by arrows. Green arrows; lumen of strongly Cb-Emerald fluorescent tubular regions lacking membranous structures. Magenta arrows; membranous structures in the lumen of tubules with low or undetectable Cb-Emerald fluorescence. All scale bars, 1 µm.

DOI: https://doi.org/10.7554/eLife.50598.032

either all parasites in a vacuole are connected by a central residual body, or not. This may suggest that the ability to produce highly organised rosettes is established during the first round of cell division when the residual body forms. This mixed population of rosetting and non-rosetting parasites, in an isogenic strain, gave us a unique opportunity to determine whether rosetting is important for efficient intravacuolar cell-cell communication. Using longCAP complemented parasites, we show that in rosetting vacuoles there is efficient RTORP with all parasites able to transfer material to the bleached parasite. Within the same clonal population, non-rosetting vacuoles displayed severe defects in the RTORP with parasites seemingly only communicating in pairs. This strongly suggests that highly organised parasite rosetting within the parasitophorous vacuole is important for rapid material transfer between cells. The majority of rosetting ΔCAP$^{longCAP}$ parasites do not contain the juxtanuclear actin polymerisation centre, suggesting this actin accumulation is not required for rosetting, as previously observed for Formin two depleted parasites. Furthermore, most ΔCAP parasites in the vacuole remained synchronised in their stage of replication (89–93%) at various timepoints (24–30 hr). This is different to other studies where mutants with defects in rosetting and RTORP between parasites show only 25–75% of synchronicity at various timepoints (24–30 hr) (*Frénal et al., 2017b*; *Tosetti et al., 2019*). Our results therefore indicate that neither RTORP between daughter cells nor the formation of a rosette are predictors for synchronicity of division.

The largely synchronous division in ΔCAP parasites could be explained by our CLEM results. Despite their disorganised appearance and loss of rapid cell-cell communication, ΔCAP parasites are still connected by a decentralised residual body. This connection, although not facilitating rapid transfer of proteins between parasites that are further apart, could allow for slow or minimal transfer of proteins which is sufficient to synchronise divisions. An alternative hypothesis could be that metabolites, not proteins, are required to synchronise divisions, and their diffusion through the decentralised residual body is quicker. Whatever the basis for synchronicity is, rosetting and rapid cell-cell communication are not essential and their analysis cannot be reliably used to predict whether connections between parasites exist. At 30 hr post-infection, when the number of parasites in the vacuole has further increased, 11% of ΔCAP parasites divide in asynchrony. Whether this is due to connections between parasites being severed/lost, or if it's a consequence of the increased distance between parasites is not known. The differing rates of synchronous division in mutants that affect rosetting may be explained by their ability to sustain connections through several rounds of division. It is a possibility that F-actin maintains tubular connections throughout divisions. To test this, other mutants with rosetting phenotypes would need to be assessed for their connectivity. In the non-chromobody expressing ΔCAP lines, we observed a continuous membranous structure

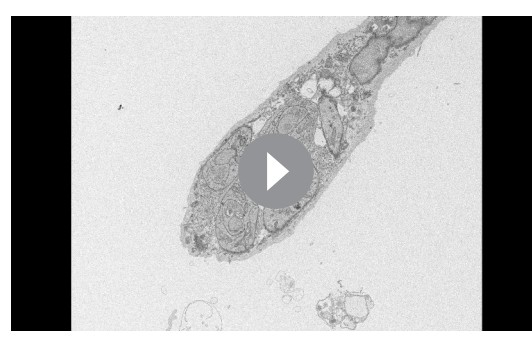

**Video 10.** SBF SEM of the Cb-EmFP ΔCAP vacuole from *Figure 9* with 3D model overlays. All SBF SEM slices of the Cb-EmFP ΔCAP vacuole from *Figure 9*. The 3D model highlights residual body tubules containing high chromobody signal and no internal membranous structures (green model) and residual body tubules with low or undetectable chromobody signal and internal membranous structures (magenta model). Parasite openings at the basal pole are also highlighted (orange ring). The volume of the SBF SEM dataset shown is indicated in the Materials and methods.

DOI: https://doi.org/10.7554/eLife.50598.033

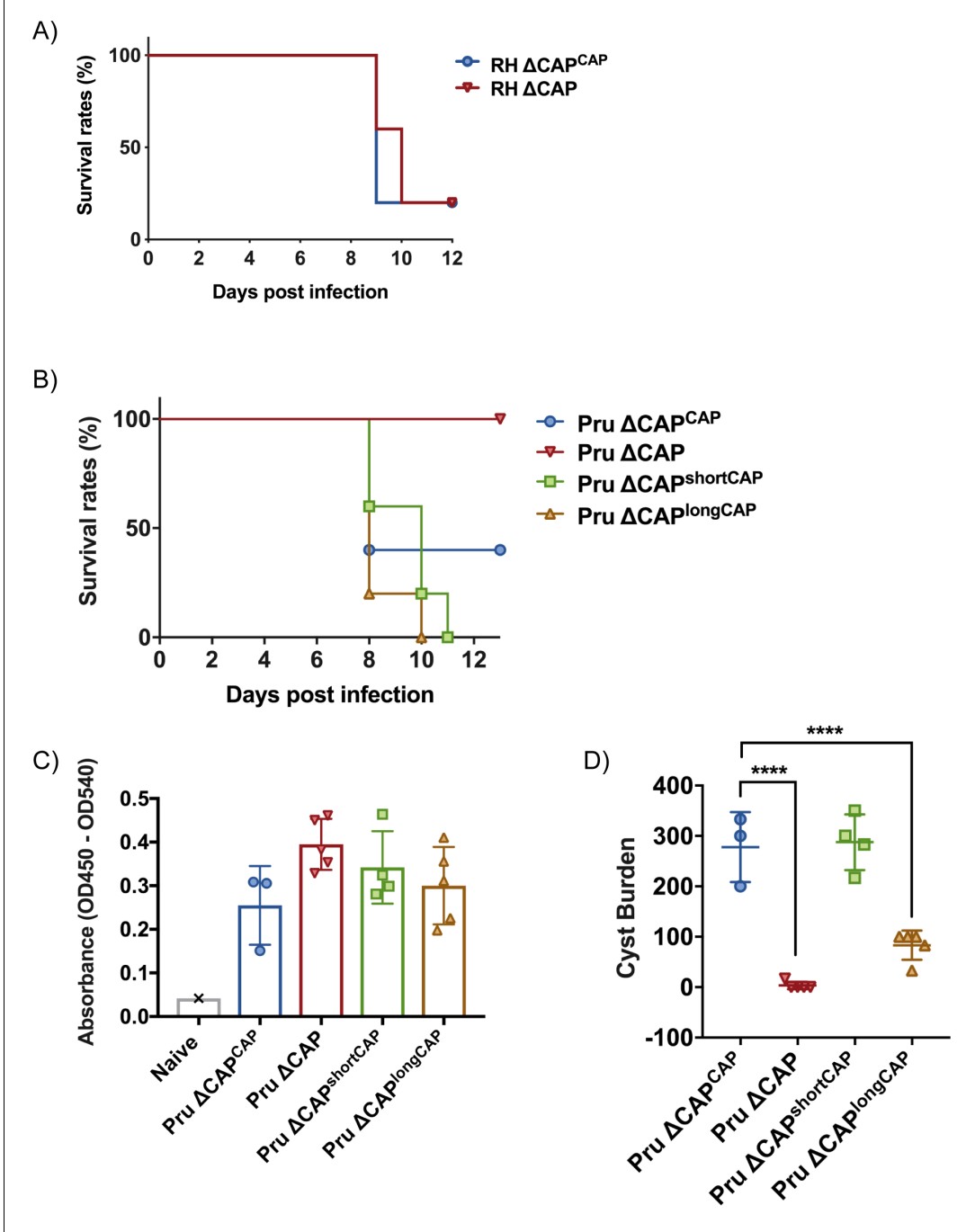

**Figure 10.** CAP is essential for virulence in type II but not type I parasites. (**A**) Survival rates of C57BL/6 mice infected with ~25 RH ΔCAP$^{CAP}$ or RH ΔCAP parasites. (**B**) Survival rates of C57BL/6 mice infected with 50,000 Pru ΔCAP parasites or with complementing CAP variants. (**C**) ELISA testing sera reactivity of naïve or *Toxoplasma* infected mice using *Toxoplasma* antigen. (**D**) Cyst burden in the brain of C57BL/6 mice 32 days post-infection with 5,000 Pru ΔCAP parasites or with complementing CAP variants. For all experiments, five animals were infected for each strain.

DOI: https://doi.org/10.7554/eLife.50598.034

The following source data is available for figure 10:

**Source data 1.** Numerical data of the graphs presented in *Figure 10A,B,C and D*.

DOI: https://doi.org/10.7554/eLife.50598.035

within the tubules connecting all parasites. We speculate that this may be the endoplasmic reticulum (ER). However, In the Cb-EmFP ΔCAP line, membranous structures were only observed in residual body tubules with low or undetectable levels of chromobody signal. In contrast, tubules with high levels of chromobody expression did not contain any membranous structures. As we imaged only single vacuoles for the chromobody and non-chromobody expressing ΔCAP lines, we cannot be certain whether the chromobody may affect tubular content or whether this is natural variation between cells. As all parasites in the chromobody expressing vacuole appear to divide in synchronicity, it is unlikely that the ER-like structure contributes to exchange of material.

The modest phenotypic defect associated with the loss of a highly conserved actin regulator in cell culture prompted us to investigate CAP function in animal infections. Despite being dispensable for type I RH parasite virulence, CAP is essential for type II Pru parasite virulence in mice, and complementation with either shortCAP or longCAP restored lethality. At lower non-lethal doses, longCAP complementation led to markedly reduced cyst formation in the brain. The underlying basis for this has not been explored here and it could be that it is the reduced fitness of the strain, rather than stage conversion phenotype, that causes a reduction of parasites reaching the brain and forming cysts.

In summary, our results support a biologically important role for CAP in *Toxoplasma*, potentially through its predicted function in regulating the monomeric actin pool. Interestingly, actin dependent processes were affected to differing extents in ΔCAP parasites, such as rosetting being completely lost while apicoplast inheritance was unaffected. This surely reflects the different spatial requirements for actin turnover within a cell. It is likely that the local concentration of actin, actin binding proteins such as the formins and the different myosins facilitate this. It is interesting to note here that CAP deletion led to only partial overlapping phenotypes with the formins, indicating that the interplay of CAP and the different formin proteins is complex. The results obtained here raise a few questions that will be interesting to study in the future. How does actin help to initiate and maintain the centralised residual body? How are ER connections between parasites in a decentralised residual body maintained and what is the function of this connection? What is the functional role of longCAP with its concentration at the apex of the parasite? Why are some actin dependent processes completely reliant on CAP while others are not? Why does loss of CAP appear to prevent F-actin formation in the juxtanuclear region of the cytosol and appear to increase F-actin formation in the residual body? This suggests that CAP might have different mechanisms of action dependent on its association with other actin binding proteins at different cellular locations. Each of these questions will require careful analysis and the cell lines described here will likely provide useful tools to investigate these in the future.

## Materials and methods

### Plasmid construction

All primers used in this study are listed in *Supplementary file 1*. All synthetic DNA used in this study is listed in *Supplementary file 2*. The key resources table is present in *Supplementary file 4*.

To generate the CAP-GFP fusion plasmid, pUPRT_CAP_GFP, the *Toxoplasma CAP* gene (TGME49_310030) 5'UTR was amplified from genomic DNA using primer pair P7/P42 and Gibson assembled (*Gibson et al., 2010*) with a synthetic *CAP* cDNA-*XmaI*-eGFP-*BamHI* sequence (GeneArt strings, Life Technologies, Massachusetts, United States) into the *BamHI* and *PacI* digested UPRT targeting vector, pUPRT-HA (*Reese et al., 2011*).

To generate the CAP cKO plasmid, pG140_CAP_cKO_LoxP111, the *CAP* 5'UTR with a loxP site inserted 111 bp upstream of the *CAP* start codon, and a recodonised *CAP* cDNA-HA sequence, were synthesised (GeneArt strings, Life Technologies). These DNA fragments were Gibson cloned (*Gibson et al., 2010*) into the parental vector p5RT70loxPKillerRedloxPYFP-HX (*Andenmatten et al., 2013*) which had been digested using *ApaI* and *PacI*, creating an intermediate plasmid. Next, the *CAP* 3'UTR was amplified from genomic DNA using primer pair P3/P4 while mCherry flanked by *GRA* gene UTRs was amplified from pTKO2c (*Caffaro et al., 2013*) using primers P5/P6. These PCR products were Gibson cloned (*Gibson et al., 2010*) into the SacI digested intermediate plasmid to create pG140_CAP_cKO_LoxP111.

To generate pUPRT_CAP, the *CAP* 5'UTR was amplified from genomic DNA using primer pair P7/P42. This DNA fragment and a synthetic *CAP* cDNA-*BamHI*-HA sequence (GeneArt strings, Life Technologies), were Gibson cloned (*Gibson et al., 2010*) into *BamHI* and *PacI* digested UPRT targeting vector pUPRT-HA (*Reese et al., 2011*).

To generate pUPRT_CAP_C6,8A, the *CAP* 5'UTR was amplified from genomic DNA using primer pair P7/P42 and, alongside a synthetic *CAP* cDNA-HA sequence with C6,8A mutations (GeneArt strings, Life Technologies), was Gibson cloned (*Gibson et al., 2010*) into *BamHI* and *PacI* digested UPRT targeting vector pUPRT-HA (*Reese et al., 2011*).

To generate pUPRT_CAP_M1L, pUPRT_CAP was amplified with primer pair P7/P8 to introduce the M1L point mutation.

To generate pUPRT_CAP_M37L, pUPRT_CAP was amplified with primer pair P9/P10 to introduce the M37L point mutation.

To generate pG140_DiCre, the plasmid containing DiCre_T2A, two synthetic DNA fragments were produced containing *Ku80* homology region-alpha-tubulin promoter-FRB-Cre2-T2A-chloramphenicol resistance cassette-T2A-FKBP-Cre1-*SAG1* 3'UTR-*Ku80* homology region (gBlock gene fragments, Integrated DNA Technologies, Iowa, United States). These DNA fragments were Gibson cloned (*Gibson et al., 2010*) into *ApaI* and *SacI* digested parental vector, p5RT70loxPKillerRedlox-PYFP-HX (*Andenmatten et al., 2013*).

To generate plasmids expressing CAS9 and a single guide RNA (sgRNA), we used plasmid pSAG1::CAS9-U6::sgUPRT as a backbone (Addgene plasmid #54467) (*Shen et al., 2014*). The plasmid was amplified with primer P11 and another containing a sgRNA to replace the UPRT-targeting sgRNA. Guide RNA sequences were selected using the Eukaryotic Pathogen CRISPR gRNA Design Tool (*Peng and Tarleton, 2015*). All sgRNA-expressing plasmids used in this study were generated by this strategy. The sgRNA-containing primers are listed in *Supplementary file 1*.

For creation of plasmids with multiple sgRNAs, first, two separate vectors, each with a sgRNA, were generated as described above using the parental vector pSAG1::CAS9-U6::sgUPRT. Then, primers P12/P13 were used to amplify one of the sgRNA regions which was Gibson cloned (*Gibson et al., 2010*) into the other *KpnI* and *XhoI* digested plasmid, creating a multiple sgRNA plasmid.

## Culturing of parasites and host cells

*T. gondii* tachyzoites were cultured in human foreskin fibroblasts (HFF) and in Dulbecco's modified Eagle's medium (DMEM) with GlutaMAX (Invitrogen, California, United States) supplemented with 10% fetal bovine serum and maintained at 37°C with 5% $CO_2$. Cultured cells and parasites were screened monthly against mycoplasma contamination by PCR using the universal mycoplasma detection kit (ATCC).

## Generation of parasite lines

To generate the conditional *CAP* knockout strain (RH DiCreΔ*ku80*Δ*hxgprt*_LoxCAP-HA, referred to here as LoxPCAP), first, the *ApaI* linearised plasmid pG140_CAP_cKO_LoxP111, carrying the *HXGPRT* cassette, was transfected into the RH DiCreΔ*ku80*Δ*hxgprt* strain (*Andenmatten et al., 2013*). Resistant parasites were cloned. Next, the DiCre conditional knockout function of the strain was restored. The linearised pG140_DiCre plasmid, carrying the chloramphenicol resistance cassette and homology with the *Ku80* UTRs, was transfected into the strain. To increase *Ku80*-specific insertion efficiency, a plasmid with multiple *Ku80*-targeting sgRNAs was generated as described above using primer pairs P11/P14 and P11/P15. Resistant parasites were cloned. Integration into the *CAP* endogenous locus was confirmed using primer pairs P16/P17 and P18/P19. Replacement of *CAP* gDNA was confirmed using primers P20/P21. Excision of the floxed *CAP* sequence was confirmed with primer pairs P16/P20.

To complement the LoxPCAP strain with CAP-expressing constructs, the linearised plasmid pUPRT_CAP, pUPRT_CAP_M1L or pUPRT_CAP_M37L was transfected alongside pSAG1::CAS9-U6::sgUPRT. 5-fluorodeoxyuridine (FUDR) resistant parasites were cloned.

*CAP* was subsequently excised from the above strains by addition of 50 nM rapamycin in DMSO for 4 hr at 37°C with 5% CO2, before washout, and excised parasites were cloned. Next, to aid with experimentation, an mCherry fluorescent construct was integrated into the *Ku80* locus, replacing the

present DiCre_T2A construct. mCherry flanked by *GRA* UTRs was amplified from pG140_CAP_cKO_-LoxP111 using primer pair P22/P23 which also carries 30 bp homology with the *Ku80* locus. To increase *Ku80*-specific insertion efficiency, a plasmid with multiple *Ku80*-targeting sgRNAs was generated as described above using primer pairs P11/P24 and P11/P25 and co-transfected with the PCR product. A population of parasites expressing mCherry were sorted by flow cytometry using a BD Influx cell sorter (BD Biosciences). Parasites were subsequently cloned, generating the strains ΔCAP, ΔCAP^CAP, ΔCAP^shortCAP and ΔCAP^longCAP.

The Pru Δ*ku80*Δ*cap* strain (referred to as Pru ΔCAP) was generated by amplifying the *HXGPRT* resistance cassette from the pG140_CAP_cKO_LoxP111 plasmid using primer pair P26/P27. To direct insertion of the PCR product to the *CAP* locus, a plasmid with multiple *CAP*-targeting sgRNAs was generated as described above using primer pairs P11/P28 and P11/P29 and co-transfected with the PCR product into the Pru Δ*ku80*Δ*hxgprt* strain (*Fox et al., 2011*). Resistant parasites were cloned. To complement the Pru Δ*ku80*Δ*cap* strain with CAP-expressing constructs, the linearised plasmid pUPRT_CAP, pUPRT_CAP_M1L or pUPRT_CAP_M37L was transfected alongside pSAG1:: CAS9-U6::sgUPRT. FUDR resistant parasites were cloned. This resulted in the Pru ΔCAP^CAP, Pru ΔCAP^shortCAP and Pru ΔCAP^longCAP strains.

To generate fluorescent Δ*gra2* parasites for FRAP experimentation, mCherry flanked by *GRA* UTRs was amplified from pG140_CAP_cKO_LoxP111 using primer pair P30/P31 with *UPRT* locus overhangs. This PCR product was co-transfected with pSAG1::CAS9-U6::sgUPRT into the RH Δ*ku80* Δ*gra2* strain (*Rommereim et al., 2016*). An FUDR resistant population was obtained.

The RH DiCre_T2A Δ*ku80*Δ*hxgprt* line was generated by integrating the DiCre construct into the *Ku80* locus in RH Δ*ku80*Δ*hxgprt* parasites (*Huynh and Carruthers, 2009*). The DiCre construct was amplified from pG140_DiCre using primer pair P32/P33 which also carries 30 bp homology with the *Ku80* locus. To increase *Ku80*-specific insertion efficiency, a plasmid with a *Ku80*-targeting sgRNA was generated as described above using primer pairs P34/P11 and co-transfected with the PCR product. Resistant parasites were cloned.

To assess the efficiency of rapamycin-dependent excision of the DiCre strains, the Killer Red gene-swap construct was amplified from p5RT70loxPKillerRedloxPYFP-HX (*Andenmatten et al., 2013*) using primer pair P35/P36. This PCR product was co-transfected with pSAG1::CAS9-U6:: sgUPRT for targeted insertion into the *UPRT* locus. FUDR resistant parasites were cloned.

To generate a CAP C-terminal endogenous HA-tagged line, a synthetic DNA repair template was produced containing *CAP* gDNA (a section of which is recodonised), a HA tag and the *CAP* 3'UTR (gBlock gene fragments, Integrated DNA Technologies). The DNA repair template was amplified using primer pair P37/P38. To direct insertion of the PCR product to the *CAP* locus, a plasmid with a single *CAP*-targeting sgRNA was generated as described above using the primer pairs P39/P11 and was co-transfected with the PCR product into RH Δ*ku80* parasites. A population of parasites expressing the co-transfected Cas9-GFP-containing plasmid were sorted by flow cytometry using a BD Influx cell sorter (BD Biosciences).~70% of these parasites expressed the HA peptide by IFA.

The actin chromobody CAP KO strain, RH Cb-Emerald Δ*ku80*Δ*cap* (Cb-EmFP ΔCAP), was generated by amplifying mCherry flanked by GRA UTRs from pG140_CAP_cKO_LoxP111 using primer pair P40/P41 with *CAP* locus overhangs. To direct insertion of the PCR product to the *CAP* locus, a plasmid with multiple *CAP*-targeting sgRNAs was generated as described above using primer pairs P11/P28 and P11/P29 and was co-transfected with the PCR product into the RH Cb-Emerald Δ*ku80* cell line (*Periz et al., 2017*). A population of parasites expressing both the mCherry and Cb-Emerald fluorophores were sorted by flow cytometry using a BD Influx cell sorter (BD Biosciences) before being cloned. To complement the RH Cb-Emerald Δ*ku80*Δ*cap* strain with CAP-expressing constructs, the linearised plasmid pUPRT_CAP, pUPRT_CAP_M1L or pUPRT_CAP_M37L was transfected alongside pSAG1::CAS9-U6::sgUPRT. FUDR resistant parasites were cloned. This resulted in the Cb-EmFP ΔCAP^CAP, Cb-EmFP ΔCAP^shortCAP and Cb-EmFP ΔCAP^longCAP strains.

## Parasite transfection and selection

To generate stable transformants, 0.5–1 × 10^7 of freshly released parasites were transfected with either 25 μg of linearised template DNA, 5 μg of linearised template DNA and 20 μg of a corresponding gRNA-specific CRISPR/CAS9 plasmid or template DNA produced from one ethanol precipitated PCR and 20 μg of a corresponding gRNA-specific CRISPR/CAS9 plasmid. Selection on the

basis of 5-fluorodeoxyuridine (20 µM), mycophenolic acid (25 µg ml$^{-1}$), xanthine (50 µg ml$^{-1}$) or chloramphenicol (21 µM) was performed according to the selection cassette used.

## Preparation of parasite genomic DNA

Genomic DNA was extracted from *T. gondii* tachyzoites to use as a PCR template by pelleting parasites and resuspending in PBS. DNA extraction was then performed using the Qiagen QIAamp DNA blood mini kit as per the manufacturer's protocol.

## IFA

Parasites were seeded onto HFFs grown on coverslips. 16–24 hr after seeding, the coverslips were fixed in 3% formaldehyde for 15 min at room temperature then permeabilised in 0.2% Triton X-100/ PBS for 3–10 min and blocked in 3% BSA/PBS for 1 hr. Staining was performed using appropriate primary antibodies and goat Alexa Fluor 488-, Alexa Fluor 594- and Alexa Fluor 647-conjugated secondary antibodies (1:2000) alongside DAPI (5 µg/ml). Coverslips were mounted on glass slides with SlowFade gold antifade mountant (Life Technologies). Antibody concentrations used were: rat anti-HA high affinity (Roche, Basel, Switzerland) (1:1000), rabbit anti-*T. gondii* CAP (1:2000), mouse anti-*Toxoplasma* antigen B1247M (Abcam) (1:1000) and rabbit anti-*T. gondii* RON4 (*Leriche and Dubremetz, 1991*) (1:2000).

Widefield images were generated with a Ti-E Nikon microscope using a 63x or 100x objective (Tokyo, Japan). Images were processed with Nikon Elements software. Confocal images were taken using a Zeiss LSM-780 inverted confocal laser scanning microscope with a 63x objective. Images were processed with Zeiss Zen Black software (Oberkochen, Germany).

## Western blot

Western blot samples were obtained by scraping and lysing intracellular parasites in 200 µl 1x Laemmli buffer (2% SDS, 10% glycerol, 5% 2-mercaptoethanol, 0.002% bromophenol blue and 125 mM Tris HCl, pH 6.8). Samples were subjected to SDS-PAGE under reducing conditions before being transferred to a nitrocellulose membrane. Immunoblotting was performed with appropriate primary antibodies in 0.1% Tween 20, 3% skimmed milk/PBS. Bound secondary fluorochrome-conjugated antibodies were visualised using the Odyssey Infrared Imaging System (LI-COR Biosciences, Nebraska, United States).

Antibody concentrations used were: rat anti-HA high affinity (Roche) (1:1000), rabbit anti-*T. gondii* CAP (1:2000), mouse anti-*Toxoplasma* [TP3] (Abcam, Cambridge, United Kingdom) (1:1000). Goat anti-mouse IRDye 800CW (LI-COR) (1:20000), Donkey anti-rabbit IRDye 680LT (LI-COR) (1:20000), Goat anti-rat IRDye 680LT (LI-COR) (1:20000).

## Generation of *T.gondii* CAP antibody

To generate the shortCAP expression plasmid, pET-28_CAP_A38toC203, a *Toxoplasma* CAP_A38-toC203 recodonised cDNA sequence was synthesised (gBlock gene fragments, Integrated DNA Technologies) and Gibson cloned (*Gibson et al., 2010*) into *BamHI* and *NdeI* digested pET-28a(+) plasmid (Merck, Darmstadt, Germany). This allowed for expression of an N-terminal 6xHis tagged shortCAP recombinant protein in *Escherichia coli* BL21 cells under the control of T7 *lac* promoter. Short CAP was expressed and His tag purified using Ni-NTA affinity purification under native conditions using the standard manufacturer's protocol (Qiagen, Hilden, Germany). The shortCAP recombinant protein was used to immunise female New Zealand white rabbits (Covalab, Cambridge, United Kingdom) for generation of polyclonal antibodies.

## Flow cytometry analysis of DiCre excision

Parasites were added to a HFF monolayer and allowed to invade for 1 hr. Then, cre recombinase-mediated recombination was induced by addition of 50 nM rapamycin in DMSO for 4 hr before washout. 22 hr after infection, parasites were syringe-released from the host cell, pelleted and washed twice in PBS. Parasites were resuspended in 0.5 ml 3% formaldehyde and fixed for 10 min. The suspension was centrifuged and the pellet washed in PBS before resuspension in PBS. To remove debris, samples were passed through a 30 µm pre-separation filter (Miltenyi Biotec, Bergisch Gladbach, Germany). 20,000 events were recorded using a BD LSR II flow cytometer (BD Biosciences

California, United States). Killer Red was excited by the 561 nm laser and detected by a 600 long pass filter and either a 582/15, 610/20 or 620/40 band pass filter. YFP was excited by the 488 nm laser and detected a 505 long pass filter and either a 525/50 or 530/30 band pass filter. For quantification, and elimination of debris, total number of Killer Red+ parasites was considered 100%. This was performed at day 0, 35 and 65 of the experiment. For each condition, three biological replicates were analysed. At least 10000 Killer Red+ events were counted for each individual time point.

## Phenotypic characterisations

### Competition Assay

mCherry-expressing ΔCAP, ΔCAP$^{CAP}$, ΔCAP$^{shortCAP}$ or ΔCAP$^{longCAP}$ parasites were mixed with non-fluorescent ΔCAP$^{CAP}$ parasites at an average ratio of 60/40. At day 0, 15 and 30 in culture, the ratio was determined by flow cytometry for two biological replicates. Parasites were syringe-released from the host cell, pelleted and washed twice in PBS. Parasites were resuspended in 0.5 ml 3% formaldehyde and fixed for 5 min. The suspension was centrifuged and the pellet resuspended in 5 μg/ml DAPI/PBS for 10 min. The pellet was washed and resuspended in PBS. To remove debris, samples were passed through a 30 μm pre-separation filter (Miltenyi Biotec). Events were recorded using a BD LSR II flow cytometer (BD Biosciences). DAPI was excited by the 355 nm laser and detected by a 450/50 band pass filter. mCherry was excited by the 561 nm laser and detected by a 600 long pass filter and a 610/20 band pass filter. To eliminate debris from the analysis, events were gated on DAPI fluorescence. The ratio of control parasites (DAPI+/mCherry-) to individual CAP complements (DAPI+/mCherry+) was calculated and normalised to the day 0 ratio. The data represent two (day 15) and three (day 30) independent experiments. At least 1500 DAPI+ events were obtained for each individual time point. The results were statistically tested with a two-way ANOVA test plus a multiple comparison Sidak's test individually comparing day 15 or day 30 means to their respective day 0 mean, in GraphPad Prism 7. The data presented are as mean ±s.d.

## Intracellular growth and rosetting assay

Parasites were harvested from a T-25 and added to a coverslip coated with a HFF monolayer. After 20 hr the coverslips were fixed with 3% formaldehyde for 15 min at room temperature. Coverslips were mounted and mCherry expression used to identify parasites. For each replicate, four random fields were imaged with a 40x objective. Counts were performed in a blinded manner in duplicate for two independent experiments. The number of parasites per vacuole was determined by counting at least 265 vacuoles per strain. The number of vacuoles that rosette was determined by looking at the eight parasite/vacuoles stage, at least 90 vacuoles per strain were counted. The results were statistically tested with a one-way ANOVA test plus a multiple comparison Dunnett's test comparing all means to the ΔCAP$^{CAP}$ mean in GraphPad Prism 7. The data presented are as mean ±s.d.

## Invasion assay

Red/green invasion assays were performed. mCherry-expressing parasites were syringe-released from the host cell in an invasion non-permissive buffer, Endo buffer (44.7 mM $K_2SO_4$, 10 mM $MgSO_4$, 106 mM sucrose, 5 mM glucose, 20 mM Tris–$H_2SO_4$, 3.5 mg/ml BSA, pH 8.2). 250 μl of $8 \times 10^5$ parasites/ml in Endo buffer were added to each well of a 24-well flat-bottom plate (Falcon), which contains a coverslip with a confluent HFF monolayer. The plates were spun at 129 x g for 1 min at 37°C to deposit parasites onto the monolayer. The Endo buffer was gently removed and replaced with invasion permissive medium (1% FBS/DMEM). These parasites were allowed to invade for 1 min at 37°C after which the monolayer was gently washed twice with PBS and fixed with 3% formaldehyde for 15 min at room temperature. Extracellular parasites were stained with mouse anti-*Toxoplasma* antigen B1247M (Abcam) 1:1000 and goat anti-mouse Alexa Fluor 488, following the IFA protocol. For each replicate, three random fields were imaged with a 40x objective. Three independent experiments were performed in duplicate. The number of intracellular (mCherry+/488-) and extracellular (mCherry+/488+) parasites was determined by counting, in a blinded fashion, at least 758 parasites per strain. The results were statistically tested with a one-way ANOVA test plus a multiple comparison Dunnett's test comparing all means to the ΔCAP$^{CAP}$ mean in GraphPad Prism 7. The data presented are as mean ± s.d.

## Egress assay

Parasites were added to a HFF monolayer, in a ibidi μ-plate 96 well, and grown for 30 hr. The wells were washed twice with PBS and the media was exchanged for 80 μl Ringers solution (155 mM NaCl, 3 mM KCl, 2 mM CaCl$_2$, 1 mM MgCl$_2$, 3 mM NaH$_2$PO$_4$, 10 mM HEPES, 10 mM glucose). To artificially induce egress, 40 μl of Ringer's solution containing 150 μM BIPPO (50 μM final conc) was added to each well. At specified time points the cells were fixed by adding 26 μl 16% formaldehyde (3% final conc) for 15 min. Cells were washed in PBS and stained with DAPI (5 μg/ml). Automated image acquisition of 25 fields per well was performed on a Cellomics Array Scan VTI HCS reader (Thermo Scientific, Massachusetts, United States) using a 20 × objective. Image analysis was performed using the Compartmental Analysis BioApplication on HCS Studio (Thermo Scientific). Egress levels were determined in triplicate for three independent assays. At least 11987 vacuoles per strain were counted at t = 0 s. Subsequent time point vacuole counts were normalised to t = 0 to determine how many vacuoles had egressed. The results were statistically tested with a two-way ANOVA test plus a multiple comparison Dunnett's test comparing all means to the ΔCAP$^{CAP}$ mean, at each time point separately, in GraphPad Prism 7. The data presented are as mean ± s.d.

## Live egress

Parasites were added to a HFF monolayer, in a ibidi μ-plate 96 well, and grown for 30 hr. The wells were washed twice with PBS and the media was exchanged for 80 μl Ringers solution (155 mM NaCl, 3 mM KCl, 2 mM CaCl$_2$, 1 mM MgCl$_2$, 3 mM NaH$_2$PO$_4$, 10 mM HEPES, 10 mM glucose). The plate was then transferred to a Ti-E Nikon microscope with a 37°C environmental chamber. To artificially induce egress, 40 μl of Ringer's solution containing 150 μM BIPPO (50 μM final conc) was added to each well after imaging had commenced. Images were captured every 1.8 s.

## Apicoplast segregation assay

Parasites were added to a HFF monolayer and grown for 20 hr before fixation with ice cold methanol for 2 min at room temperature. IFAs were performed using a streptavidin-Alexa Fluor 594 conjugate (Invitrogen) as a marker for the apicoplast. Correct apicoplast segregation was determined in duplicate for three independent assays. At least 228 vacuoles were counted per strain. Counts were performed in a blinded manner. The results were statistically tested with an unpaired t test in GraphPad Prism 7. The data presented are as mean ± s.d.

## Synchronicity of division and daughter cell orientation assay

Parasites were added to a HFF monolayer and grown for 20, 24 or 30 hr before fixation with 3% formaldehyde for 15 min at room temperature. IFAs were performed using rat anti-IMC3 antibodies. To determine the synchronicity of cell division within the vacuoles, anti-IMC3 staining was used to evaluate the stage of daughter cell development. Vacuoles were scored as synchronous if all daughter cells were at the same stage of development. For the 20 hr time point, vacuoles were counted blind in triplicate for three independent experiments. For the 24 and 30 hr time points, vacuoles were counted blind in duplicate for three independent experiments. At least 233 vacuoles were counted per strain, for each time-point. The results were statistically tested with an unpaired t test in GraphPad Prism 7. The data presented are as mean ± s.d. Daughter cell orientation was quantified in triplicate for three independent experiments using the 20 hr time point. At least 312 mother cells were counted blind, per strain. The results were statistically tested with a one-way ANOVA test plus a multiple comparison Dunnett's test comparing all means to the ΔCAP$^{CAP}$ mean in GraphPad Prism 7. The data presented are as mean ± s.d.

## 3D motility assay

Motility assays were performed as previously described (*Leung et al., 2014*), with minor modifications. Parasites were syringe-released from a HFF monolayer (one heavily infected T75 flask per strain) by passing through a 27 gauge needle and filtering through a 3 μm Nuclepore filter. Parasites were then centrifuged (1000 x g for 2 min) and resuspended in 40 μl motility media supplemented with 0.3 mg/mL Hoescht 33342. Matrigel was thawed on ice to prevent polymerisation and combined with parasites and motility media in a ratio of 3:1:3 respectively. Pitta chambers were perfused with 10 μL of this suspension and incubated at 27°C for 7 min on a thermoplate. Chambers were

incubated in the heated (35°C + /- 1°C) microscope enclosure for 3 min prior to imaging. Parasite nuclei were imaged, using a 20x objective, capturing 61–63 stacks of 41 z-slices 1 μm apart. To ensure conditions remained constant between parasite lines, samples used for capture were alternated. Datasets were exported to Imaris x64 v9.2.1. Using the ImarisTrack module parasites were tracked in a region of interest from which 1 μm has been cropped from the x and y edges to eliminate edge artefacts. Parasites were identified, after background subtraction, using spot detection with estimated size of 4 μm (xy) and 8 μm (z). Spots were filtered to exclude all that had duration of less than 3 s to minimise tracking artifacts. An autoregressive motion tracking algorithm was applied with a maximum distance of 6 μm and a maximum gap size of 3. Datasets were manually inspected to ensure appropriate tracking and to remove artifacts and trajectories tracked from multiple identified spots. Percent moving was calculated as the number of trajectories (>2 μm displacement)/number of objects in frame 3 (>3 s duration). Trajectory parameters were extracted directly from Imaris software. Data shown are derived from four independent biological replicates, each consisting of a minimum of two technical replicates. For all analysis, means and standard deviation were calculated for four independent biological replicates before statistical analysis using an unpaired t-test in GraphPad Prism 7. For the three strain experiment, $\Delta CAP^{shortCAP}$ and $\Delta CAP^{longCAP}$ were each compared to $\Delta CAP^{CAP}$. All technical replicates are presented in the figures, together with their mean ± s.d.

## Dense granule trafficking assay

For each condition $1 \times 10^7$ parasites were transiently transfected with 30 μg of pTub SAG1ΔGPI-mCherry plasmid (*Heaslip et al., 2016*) and immediately added to a confluent HFF monolayer in a Mattek 35 mm dish, coverslip 1.5. 12–15 hr after infection the monolayers were washed three times with pre-warmed Gibco Fluorobrite DMEM supplemented with 4% Fetal Bovine Serum. The coverslips were immediately used for imaging. The acquisitions were made with an Olympus IX71 coupled to a DELTAVISION Elite imaging system in a 37°C environmental chamber. The acquisition for each condition were made sequentially from the washing step to the acquisition with a random order to avoid any artifactual data. The acquisition analysis was made with Fiji and the MTrackJ plugin. The data represent three independent experiments. At least 42 parasites and 273 direct runs were counted per strain. The results were statistically tested with a one-way ANOVA test plus a multiple comparison Dunnett's test comparing all means to the $\Delta CAP^{CAP}$ mean in GraphPad Prism 7. The data presented are as mean ± s.d.

## FRAP

Parasites were inoculated on a confluent layer of HFFs 20 hr before experiments were performed using a Zeiss LSM-780 inverted confocal laser scanning microscope at 37°C. Acquisition and processing were performed with the Zeiss Zen Black software. Images were taken for 2 min (one image per second). Three pre-bleach images were recorded before the region of interest was photobleached ten times with a 561 nm laser at 100% power. Fluorescence intensity is presented as a percentage relative to the same area pre-bleach. These normalised intensity values were also used for the calculations below. To calculate percentage of fluorescence recovery, the final reading (116 s post-bleach) was subtracted from the reading immediately post-bleach, t = 3 s, to give percentage recovery after 116 s. To calculate percentage loss of fluorescence, the reading immediately post-bleach, t = 3 s, was subtracted from the final reading (116 s post-bleach) to give percentage loss after 116 s. These percentages were used to generate the heat map. At least 11 vacuoles were counted per strain across at least two independent experiments. The results were statistically tested with a one-way ANOVA test plus a multiple comparison Dunnett's test comparing all means to the $\Delta CAP^{CAP}$ mean in GraphPad Prism 7. The data presented are as mean ± s.d. To assess recovery type, at least 11 vacuoles were counted per strain across at least two independent experiments. The results were statistically tested with a Chi-square in GraphPad Prism 7.

## Tape unroofing SEM

Parasites were inoculated on a confluent HFF monolayer 24 hr before fixation in EM fixative (2.5% gluteraldehyde, 4% formaldehyde in 0.1 M phosphate buffer) for 30 min. Cells were washed in 0.1 M phosphate buffer (PB) and stored in 1% formaldehyde in PB at 4°C. Cells were then washed in PB at

room temperature, then washed in ddH$_2$O at RT. The cells were dehydrated stepwise from 70% to 100% ethanol before critical point drying from acetone in a CPD300 (Leica Microsystems, Vienna, Austria). After drying, the coverslips were mounted on stubs, and the HFF cells were unroofed by placing Scotch tape on the coverslips and gently peeling it off, exposing the host cytoplasm and the parasitophorous vacuoles. The cells were coated with 7 nm platinum in a Q150R Sputter Coater (Quorum Tech, East Sussex, UK) before viewing in a Phenom ProX SEM (Thermo Scientific) at 10 kV, 1024 × 1024 pixel frame, on 'high' quality.

## FIB SEM

Immediately following FRAP experimentation, as described above, parasites were fixed in 4% form-aldehyde for 15 min at 37°C before washing in 0.1 M PB. Cells were then fixed in EM fixative (2.5% gluteraldehyde, 4% formaldehyde in 0.1 M PB) for 30 min at room temperature. Cells were washed in 0.1 M PB and stored in 1% formaldehyde in 0.1 M PB. After fixation, samples were transferred to a Pelco BioWave Pro+ microwave (Ted Pella) for processing using a protocol adapted from the NCMIR protocol (*Deerinck et al., 2010*). See *Supplementary file 3* for full BioWave program details. The SteadyTemp plate was set to 21°C unless otherwise indicated. Each step was performed in the microwave, except for the buffer and ddH$_2$O wash steps, which consisted of two washes on the bench and two washes in the microwave (250 W for 40 s). The cells were washed (as above) in 0.1 M PB, stained with 2% osmium tetroxide and 1.5% potassium ferricyanide (v/v) for 14 min under vacuum (with/without 100 W power at 2 min intervals), and then washed in ddH$_2$O (as above). Next, the cells were incubated in 1% thiocarbohydrazide in ddH$_2$O (w/v) (14 min,vacuum, 100 W on/off at 2 min intervals) with SteadyTemp plate set to 40°C, followed by ddH$_2$O washes (as above), and then a further stain with 2% osmium tetroxide in ddH$_2$O (w/v) (14 min,vacuum, 100 W on/off at 2 min intervals), followed by ddH$_2$O washes (as above). The cells were then incubated in 1% aqueous ura-nyl acetate (vacuum, 100 W on/off at 2 min intervals, SteadyTemp 40°C), and then washed in ddH$_2$O (as above, except with SteadyTemp at 40°C). Walton's lead aspartate was then applied (vacuum, 100 W on/off at 2 min intervals, SteadyTemp 50°C), and the cells were washed (as above) and dehydrated in a graded ethanol series (70%, 90%, and 100%, twice each), at 250 W for 40 s without vacuum. Exchange into Durcupan ACM resin (Sigma-Aldrich, Missouri, United States) was performed in 50% resin in ethanol, at 250 W for 3 min, with vacuum cycling (on/off at 30 s intervals), and then pure Durcupan was infiltrated in four microwave steps with the same settings, before embedding at 60°C for 48 hr.

Focused ion beam scanning electron microscopy (FIB SEM) data was collected using a Crossbeam 540 FIB SEM with Atlas 5 for 3-dimensional tomography acquisition (Zeiss). Segments of the cell monolayer containing the cells of interest were trimmed, polished with a diamond knife (removing uneven resin at the base of the monolayer to provide a flat surface for tracking marks), mounted on a standard 12.7 mm SEM stub using conductive epoxy (ITW Chemtronics), and coated with a 5 nm layer of platinum.

The specific cells of interest were relocated by imaging through the platinum coating at an accel-erating voltage of 20 kV and correlating to previously acquired fluorescence microscopy images. After preparation for milling and tracking, images were acquired at 5 nm isotropic resolution throughout each region of interest, using a 10 μs dwell time. During acquisition the SEM was oper-ated at an accelerating voltage of 1.5 kV with 1 nA current. The EsB detector was used with a grid voltage of 1,200 V. Ion beam milling was performed at an accelerating voltage of 30 kV and current of 700 pA.

After cropping to the specific region of interest comprising the entire extent of the parasitopho-rous vacuole (ΔCAP$^{CAP}$; 3080 × 634×2509 pixels; 15.4 × 3.17 × 12.545 μm; ΔCAP; 4065 × 1136×4490 pixels; 20.325 × 5.68 × 22.45 μm) and aligning the dataset (gradient align; Atlas 5), the images were processed to supress noise and slightly enhance sharpness (gaussian blur 0.75 radius, followed by unsharp mask radius 1, strength 0.6; Fiji) prior to reorienting and reslicing in the YZ plane (assigned as the XY plane for segmentation), and scaling to 10 nm isotropic resolution for seg-mentation and display.

Selected structures were segmented manually from the FIB SEM datasets and 3D reconstructions were made using the 3dmod program of IMOD (*Kremer et al., 1996*). The normal and decentralised residual body structures were manually skeletonised by tracing the approximate central axis of the structure from the posterior pore of each tachyzoite using open contours. Thus, the green skeleton

follows the lumen of the connections between parasites. The minimum number of points were placed that would still ensure the contour remained at the central axis of the volume. Where the structure branched, points linking contours from multiple extensions were placed at the approximate centre of the branch point volume. The contours were then rendered as a 50 or 100 nm tube to aid visualisation (tube diameter chosen depending on view). This model was then inspected in the X, Y, and Z-planes and corrections made to ensure the skeleton followed the approximate centre through the volume. Since not all RB-like structure extensions ended at the posterior pole of a tachyzoite, the posterior poles were highlighted by segmenting them with open contours with points at every 40 nm in Z, following the edge of the cytosol (the ribosome-containing electron lucent space where it meets the intermediate electron density surrounding the inner membrane complex), and meshing the contour as a 50 nm tube. A selected region of the putative ER lipid bilayer outer leaflet was segmented with closed contours drawn every 5–20 nm in Z (smaller Z intervals where needed to capture fenestrations/complexity); from an approximately square region around the inner face of two of the basal pores up to an arbitrary point along the decentralised residual body structure. Contour gaps were placed at the edge of this region. Two tachyzoites were also highlighted by partial coarse segmentation; closed contours drawn every 250 nm in Z through the main body of the cell (segmentation of the complex top and bottom of the tachyzoites was omitted for clarity).

## SFB SEM

Following fluorescence imaging, parasites were fixed and embedded in resin as described for FIB SEM. The area of interest was then trimmed, mounted, and approached for SBF SEM as described in *Russell et al. (2017)*. SBF SEM images were collected using a 3View 2XP system (Gatan Inc) with charge compensation, mounted on a Sigma VP scanning electron microscope (Zeiss). Images were collected at 2.1 kV with a 30 μm aperture at high vacuum with a progressive 50–65% gas charge compensation, and a 2 μs dwell time. The dataset was $32.12 \times 32.12 \times 3.4$ μm in xyz, consisting of 68 serial images of 50 nm thickness and a pixel size of $7.8 \times 7.8$ nm. The total volume of the dataset was ~3508 $\mu m^3$.

Selected structures were segmented manually from the SBF SEM dataset and 3D reconstructions were made using the 3dmod program of IMOD (*Kremer et al., 1996*), essentially as for the FIB SEM datasets. Briefly, the approximate central axes of the decentralised residual body structures were manually skeletonised from the posterior pore of each tachyzoite using open contours and rendered as 100 nm tubes. These structures were segmented with reference to the chromobody fluorescence dataset rendered in three dimensions using the ClearVolume Fiji plugin (*Royer et al., 2015*); those tubes whose path in three dimensions resembled the path of strong green fluorescent signal were rendered as green 100 nm tubes, while magenta was used for tubes whose path followed areas of low green fluorescent signal or where fluorescence was undetectable. Posterior pores were segmented as for the FIB SEM datasets, except points were placed at 50 nm intervals in Z. Segmentation was done on unprocessed SBF SEM images, while figures were prepared after denoising SBF SEM images with a 0.5 sigma Gaussian blur in Fiji. Since both LM and EM datasets lacked point landmarks for correlative registration, orthogonal views of the datasets were coarsely registered by eye in Adobe Illustrator; the chromobody fluorescence was aligned with the decentralised residual body skeleton model by linear adjustments of size and position only (no rotation and non-linear alterations).

## Animals

C57BL/6 (wild type) mice were bred and housed under pathogen-free conditions in the biological research facility at the Francis Crick Institute in accordance with the Home Office UK Animals (Scientific Procedures) Act 1986. All work was approved by the UK Home Office (project license PDE274B7D), the Francis Crick Institute Ethical Review Panel, and conforms to European Union directive 2010/63/EU. All mice used in this study were male and between 7- to 9 week old.

Mice were infected with *T. gondii* tachyzoites by intraperitoneal injection (i.p.) with either 25, $5 \times 10^3$ parasites (cyst formation) or $5 \times 10^4$ parasites (survival) in 200 μl medium on day 0. Mice were monitored and weighed regularly for the duration of the experiments.

For serum samples, mice were euthanised and blood collected into blood serum collection tubes (SAI, Infusion technologies) by puncturing the jugular vein. Blood was allowed to clot at room

temperature for 30 min, before tubes were centrifuged at 1500 x g for 10 min. Serum was collected and stored at −20°C until analysis.

### *Toxoplasma* serum antibody ELISA

*Toxoplasma* soluble antigens were extracted as previously described (*Silva et al., 2007*). In short, parasites were syringe-released from the host cell, washed once with PBS, and adjusted to $1 \times 10^8$ tachyzoites/ml with PBS containing protease inhibitors (cOmplete mini, Roche). Parasites were lysed by five freeze-thaw cycles (liquid nitrogen/37°C), followed by ultrasound sonication on ice (five 60 Hz cycle for 1 min each). Samples were centrifuged at 10,000 x g for 30 min at 4°C before supernatants were collected and protein content was determined using the BCA Protein Assay Kit (Pierce, Thermo Fisher Scientific) following the manufacturer's instructions.

To detect *Toxoplasma* antibodies in murine serum samples, 96-well plates (flat bottom, high-binding) were coated overnight with 2 µg/ml *Toxoplasma* soluble antigens at 4°C. Plates were washed with PBS/0.05% Tween-20 (v/v) (PBS-T) before blocked with 1% BSA (w/v) in PBS for 2 hr at room temperature. Bound antigens were incubated with murine sera diluted 1/10 in 1% BSA/PBS for 2 hr at room temperature, washed three times with PBS-T and bound antibodies detected by incubation for 2 hr at room temperature with anti-mouse Immunoglobulins (HRP conjugate, Darko) diluted 1/1000 in 1% BSA/PBS. Finally, plates were washed three times with PBS-T and developed by adding TMB substrate solution (Thermo Fisher Scientific). The TMB reaction was stopped by adding 2 N sulphuric acid and the absorbance measured ($OD_{450}$ minus $OD_{540}$ wave length correction) using the VersaMax Microplate Reader with SoftMax Pro Software.

### Mouse brain collection and preparation for cyst counting

To determine the number of cysts in the brain of infected animals, mice were euthanised and the brain extracted from the skull. The brain was homogenised in 1 ml PBS and stained with Rhodamine-conjugated *Dolichos biflorus* agglutinin (1/1000; Vector Laboratories) for 1 hr at room temperature. Fluorescently labelled cysts were counted using a Ti-E Nikon microscope.

### Statistical analysis

Statistical tests used are stated in individual sections above. *P*-values significance thresholds were set at: ****$p<0.0001$, ***$p<0.001$, **$p<0.01$ and *$p<0.05$. All significant results are labelled with a line and asterisk(s) in the graphs.

## Acknowledgements

We thank Marc-Jan Gubbels (Boston College) for gifting the IMC3 antibody, Jean François Dubremetz for the RON4 antibody, Markus Meissner (University of Glasgow) for the pG140 plasmid, Michael Reese (University of Texas Southwestern Medical Center) for the pUPRT-HA plasmid and David Sibley (Washington University) for the CRISPR/Cas9 plasmid. Thanks to Caia Dominicus (Francis Crick Institute) and Kaiser Hunt for critically reading the manuscript and Leandro Lemgruber (University of Glasgow) for his advice on tape unroofing SEM. We also thank the following members of science technology platforms at the Francis Crick Institute for their support: Matt Renshaw (light microscopy), Michael Howell (high throughput screening), Damian Carragher, Rhys Hefin, Phil Hobson and Graham Preece (flow cytometry), the Cell Services and Peptide Chemistry teams. This work was supported by awards to MT by The Francis Crick Institute (https://www.crick.ac.uk/), which receives its core funding from Cancer Research UK (FC001189; https://www.cancerresearchuk.org), the UK Medical Research Council (FC001189; https://www.mrc.ac.uk/) and the Wellcome Trust (FC001189; https://wellcome.ac.uk/). This work was supported by US Public Health Service grants AI137767 and AI139201 to GEW and National Institutes of Health grant awarded to Aoife Heaslip (AI1218885).

## Additional information

### Funding

| Funder | Grant reference number | Author |
|---|---|---|
| Francis Crick Institute | FC001189 | Alex Hunt<br>Jeanette Wagener<br>Lucy Collinson<br>Moritz Treeck |
| NIH | AI1218885 | Romain Carmeille<br>Aoife Heaslip |
| NIH | AI139201 | Robyn Kent<br>Gary E Ward |
| Francis Crick Institute | FC001999 | Matthew Robert Geoffrey Russell<br>Christopher J Peddie<br>Lucy Collinson |
| NIH | AI137767 | Robyn Kent<br>Gary E Ward |

The funders had no role in study design, data collection and interpretation, or the decision to submit the work for publication.

### Author contributions

Alex Hunt, Conceptualization, Formal analysis, Investigation, Visualization, Methodology, Writing—original draft, Writing—review and editing; Matthew Robert Geoffrey Russell, Jeanette Wagener, Robyn Kent, Formal analysis, Investigation, Visualization, Writing—original draft, Writing—review and editing; Romain Carmeille, Formal analysis, Investigation; Christopher J Peddie, Formal analysis, Investigation, Writing—review and editing; Lucy Collinson, Formal analysis, Supervision, Writing—review and editing; Aoife Heaslip, Formal analysis, Supervision, Visualization, Writing—original draft, Writing—review and editing; Gary E Ward, Formal analysis, Supervision, Writing—original draft, Writing—review and editing; Moritz Treeck, Conceptualization, Formal analysis, Supervision, Funding acquisition, Writing—original draft, Project administration, Writing—review and editing

### Author ORCIDs

Alex Hunt ⓘD https://orcid.org/0000-0001-7431-7156
Matthew Robert Geoffrey Russell ⓘD https://orcid.org/0000-0003-4608-7669
Jeanette Wagener ⓘD http://orcid.org/0000-0002-7227-4348
Robyn Kent ⓘD https://orcid.org/0000-0002-2584-4694
Gary E Ward ⓘD https://orcid.org/0000-0003-4138-3055
Moritz Treeck ⓘD https://orcid.org/0000-0002-9727-6657

### Ethics

Animal experimentation: All experiments were performed in accordance with UK Home Office regulations (PPL 80/2616) and approved by the ethical review panel at the Francis Crick Institute.

### Decision letter and Author response

Decision letter https://doi.org/10.7554/eLife.50598.050
Author response https://doi.org/10.7554/eLife.50598.051

## Additional files

### Supplementary files

• Supplementary file 1. Primers listed in the Materials and methods.
DOI: https://doi.org/10.7554/eLife.50598.036
• Supplementary file 2. Synthetic DNA listed in the Materials and methods.

DOI: https://doi.org/10.7554/eLife.50598.037

• Supplementary file 3. BioWave program details for FIB SEM.
DOI: https://doi.org/10.7554/eLife.50598.038

• Supplementary file 4. Key resources table.
DOI: https://doi.org/10.7554/eLife.50598.039

• Transparent reporting form
DOI: https://doi.org/10.7554/eLife.50598.040

## Data availability

All data generated or analysed during this study are included in the manuscript and supporting files. Source data files have been provided for Figures 4, 5, 6, 8 and 10. Raw data for FIB SEM supporting movies have been uploaded to EMPIAR and are available under the accession numbers EMPIAR-10324, EMPIAR-10325, EMPIAR-10326 and EMPIAR-10327.

The following datasets were generated:

| Author(s) | Year | Dataset title | Dataset URL | Database and Identifier |
|---|---|---|---|---|
| Alex Hunt, Matthew Robert Geoffrey Russell, Jeanette Wagener, Robyn Kent, Romain Carmeille, Christopher J Peddie, Lucy Collinson, Aoife Heaslip, Gary E Ward, Moritz Treeck | 2019 | Processed FIB SEM images of a parasitophorous vacuole containing Toxoplasma gondii ΔCAP parasites | https://www.ebi.ac.uk/pdbe/emdb/empiar/entry/10324/ | EMPIAR, EMPIAR-10324 |
| Alex Hunt, Matthew Robert Geoffrey Russell, Jeanette Wagener, Robyn Kent, Romain Carmeille, Christopher J Peddie, Lucy Collinson, Aoife Heaslip, Gary E Ward, Moritz Treeck | 2019 | Raw FIB SEM images of a parasitophorous vacuole containing Toxoplasma gondii ΔCAP parasites, complemented with CAP | https://www.ebi.ac.uk/pdbe/emdb/empiar/entry/10325/ | EMPIAR, EMPIAR-10325 |
| Alex Hunt, Matthew Robert Geoffrey Russell, Jeanette Wagener, Robyn Kent, Romain Carmeille, Christopher J Peddie, Lucy Collinson, Aoife Heaslip, Gary E Ward, Moritz Treeck | 2019 | Raw FIB SEM images of a parasitophorous vacuole containing Toxoplasma gondii ΔCAP parasites | https://www.ebi.ac.uk/pdbe/emdb/empiar/entry/10326/ | EMPIAR, EMPIAR-10326 |
| Alex Hunt, Matthew Robert Geoffrey Russell, Jeanette Wagener, Robyn Kent, Romain Carmeille, Christopher J Peddie, Lucy Collinson, Aoife Heaslip, Gary E Ward, Moritz Treeck | 2019 | Processed FIB SEM images of a parasitophorous vacuole containing Toxoplasma gondii ΔCAP parasites, complemented with CAP | https://www.ebi.ac.uk/pdbe/emdb/empiar/entry/10327/ | EMPIAR, EMPIAR-10327 |

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
