## [Decision Letter]

[Editors’ note: a previous version of this study was rejected after peer review, but the authors submitted for reconsideration. The first decision letter after peer review is shown below.]

Thank you for submitting your work entitled "Differential requirements of cyclase associated protein for actin turnover during the lytic cycle of *Toxoplasma gondii*" for consideration by *eLife*. Your article has been reviewed by three peer reviewers, and the evaluation has been overseen by a Reviewing Editor and a Senior Editor. The following individual involved in review of your submission has agreed to reveal their identity: Markus Meissner (Reviewer #1).

Our decision has been reached after consultation between the reviewers. Based on these discussions and the individual reviews below, we regret to inform you that your work will not be considered further for publication in *eLife* at the current time.

However, as you will see the reviewers each praised the high quality of the work and would be all happy to look at a revised manuscript. The consultation session found that it would be pertinent to perform the actin chromobody experiments to investigate directly the dynamics of actin in the parasite (suggested by reviewer 1 and 2 and reviewer 3 agreed). For other, in my view mostly doable, but of course time consuming suggestions please see the individual reviews appended below.

Reviewer #1:

In the study Hunt et al., the authors perform a careful phenotypic characterisation of parasite mutants for the actin capping protein of the apicomplexan parasite *Toxoplasma gondii*, TgCAP. Using state of the art and cutting edge technologies the authors demonstrate that CAP is alternatively translated, localised and that it's deletion causes effects in parasite growth, probably caused by defects in host cell invasion and egress. Furthermore, deletion of CAP causes alteration of the residual body, a structure connecting parasites within the parasitophorous vacuole and consequently to inefficient parasite connectivity and material transfer. In contrast, other actin dependent processes, such as apicoplast segregation or dense granule motility appear to be not or only minimally affected in absence of CAP, leading the authors to speculate that CAP is only required in the regulation of certain actin-dependent processes.

The authors present a tour-de-force to analyse the phenotype caused by deletion of cap focussing on processes, previously described to be dependent on parasite actin and/or myosins. The variety of techniques, from FIB SEM, SEM, FRAP, mouse challenge, FACs, western blot and live 3D motility assay, gives a great overview of the phenotypic consequences caused by deletion of CAP. While this characterisation is very well performed, it does not directly demonstrate a role of CAP in F-Actin dynamics in the parasite. It is now possible to measure and visualise F-actin dynamics in apicomplexan parasites, which allowed to demonstrate the role of F-actin in material transfer (Periz et al., 2017) and the independent functions of apicomplexan Formins in F-actin nucleation (see Tosetti et al., 2019). While the interpretation of the authors is certainly likely, no direct functional data regarding the role of CAP for the regulation of F-actin dynamics is provided and the conclusions are drawn based on the phenotypes obtained for other F-actin regulatory proteins. Therefore, in their resubmission, the authors should directly analyse the role of CAP in F-actin dynamics.

1) This study is about a major actin modulating protein and does not look at actin dynamics directly. However, this is now possible using the F-actin nanobody and should be done. It would be of great interest, how short CAP, long CAP and ∆CAP influence actin dynamics during intracellular replication, egress, gliding motility and invasion.

2) The authors did not look at the dynamics of CAP during these processes, which would be relatively straight forward, given that parasites expressing GFP-tagged versions are already available.

Especially the dynamics of apically localized CAP would be interesting to analyse in more detail, since it might suggest an association with Formin1, whereas the cytoplasmic CAP might be more associated with Formin2/3. This might also reflect the different effects in the complementation assays.

This should at least be discussed in more detail in a resubmission.

3) The rosetting phenotype as well as the RTORP are very interesting findings. It is great to do CLEM on FRAP / FIB imaging, which to our knowledge is the first time this has been done in *Toxoplasma*. It is technically extremely challenging, as is indicated by the lack of statistical analysis for the FIB data. However, the discussion of the part of the analysis is a bit biased. One could immediately question if the correct time scale was analyzed for the FRAP analysis in those parasites lacking a central residual body. A long and tubular cytoplasmic bridge connecting two parasites might result in much longer diffusion times for cytoplasmic proteins. Also, as actin was not visualized, more prominent actin filaments (caused by deletion of CAP) within the tubular residual body might delay diffusion of mCherry.

Similar results as the FIB SEM could have been generated in a quantifiable manner by expressing the actin-binding chromobody (depending on the localization within the various CAP mutants) or a strong fluorescent reporter expressed at the plasma membrane.

Reviewer #2:

Actin plays a critical role in the lytic cycle of apicomplexan parasites, and is known to be important for motility, organelle inheritance, and cell-cell communication. This manuscript explores the role of the putative actin regulator cyclase-associated protein (TgCAP) in a variety of such actin-dependent processes. A methodological improvement introduced in this manuscript is the stable DiCre construct, which retains its excision capacity over long periods of time.

The authors provide evidence for two protein isoforms of TgCAP that arise from two translation start sites and result in a cytosolic short isoform and a long isoform that is apically localized through palmitoylation sties not present in the short isoform. While effects on a number of different processes are measured for TgCAP mutants, the authors never directly observe how or whether actin turnover is affected, nor do they explore how the distinct localizations contribute to the role of each isoform.

Previous studies (Frénal et al., 2017, Tosetti et al., 2019) have observed asynchronous division in parasites with defects in intravacuolar cell-cell communication, upon disruption of F-actin and a specific myosin. One contrasting finding in the current manuscript is that loss of TgCAP can decouple these two phenotypes: reporter proteins do not readily diffuse between parasites in the entire vacuole, yet cell division remains synchronous. As detailed below, more work is needed to determine whether the TgCAP phenotype is truly distinct from the previously reported phenotypes. However, based on the previous reports involving actin and myosin in intravacuolar contact/communication, and the lack of mechanistic information regarding TgCAP, the current study does not significantly advance our understanding of the role of actin in the *Toxoplasma* lytic cycle.

1) This manuscript focuses on phenotypic assessment of processes for which actin is important, but there are no direct observations of the effects of TgCAP on actin turnover, as is suggested in the title. This could be addressed by using modified actin-chromobodies, which have been employed to study actin dynamics in *Toxoplasma* (Periz et al., 2017), to visualize filamentous actin throughout the cells and networks of cells in the various TgCAP mutants.

2) In all the phenotypic assays, the short isoform of TgCAP is able to complement the knockout as well as or better than the long isoform of TgCAP. While the authors provide compelling evidence that the two isoforms have distinct cellular localizations, there is no clear evidence that the localization is playing a critical role. One explanation for the more severe defects in complementation observed with the long isoform is that it is expressed at substantially lower levels relative to the short isoform (Figure 3F), and that overall levels of TgCAP are critical for its function. A prediction of this would be that a palmitoylation mutant (C6,8A) that could only be expressed as the long isoform (M37L mutant) would be unable to rescue the phenotypes as well as the short isoform.

3) An important finding in this manuscript is that while TgCAP mutants have defects in protein diffusion between all parasites within a vacuole, they are still able to divide synchronously. Parasite division synchronicity assays presented in this manuscript are performed at 20 hours post infection. Frénal et al., showed increased severity in defects in synchronized division of myosin knockouts that occurred from 24 to 30 hours post infection. To ensure that the lack of synchronization defects observed in this manuscript is not simply due to earlier measurements, the authors should measure whether TgCAP mutants are still able to maintain synchronous divisions at these slightly later timepoints when there might be differences compared to wildtype.

4) If levels of TgCAP are important for function, it is important to understand how the levels in the complemented strains compare to wildtype cells. This can be measured using the TgCAP antibody presented in this manuscript.

Reviewer #3:

You've asked me to read this manuscript from the point of view of an actin biologist. And I'm afraid from that point of view this manuscript is not very enlightening – while it confirms that CAP is physiologically important, it can't pin down a function for the long form of CAP, makes no comment on what the CAP is doing to the actin, and all the data presented describe the behaviour of CAP alone in some detail but nothing of actin or actin regulatory proteins.

From a reviewer's and editor's point of view, this leads to three conclusions:

a) I'm really not competent to say whether the advances in apicomplexan biology are significant or not. They seem to be important, but I defer to reviewers with expertise in this area.

b) However, the authors' attempts at justifying the results in terms of actin biology are not supported. There is no evidence that the CAP is having any effect on *Toxoplasma's* actin turnover. It should therefore be removed from the title. The evidence is only inferred from other systems. Additionally, since the stability of parasite actin is completely different (cf the excellent PNAS paper from Raunser's lab) even this inference is not trustworthy.

Similarly, the Discussion needs to be rewritten. It is not true that "our results strongly support an actin regulatory role for CAP in *Toxoplasma*". The results strongly support the biological importance of *Toxoplasma* CAP; the correlation with the biological importance of actin is very patchy. Again, the actin regulatory role is inferred from other work rather than shown here. The least indirect connection with actin is made via dense granule transport; however the MBoC paper that connects the two is already indirect (measuring loss of migration in the presence of strong actin inhibitors like cytochalasin and promoters like jasplakinolide, which of course cause secondary effects) – it is certainly not sound to conclude that a mutation that causes an opposite phenotype to actin inhibitors (dCAP mutant granules move fast) must solely be acting through actin dynamics.

c) if this sounds very negative about the paper as a whole, it's not meant to be. The data seem clear and important. I'm not sure whether they belong in a general biology journal like *eLife* – perhaps that's the reason why the actin angle has been pushed harder than the data support. I would recommend – first, recast this as a paper on apicomplexan biology, with the Discussion raising the interesting possibility (yet again) that parasite actin has a cryptic biology that needs to be understood, rather than stating that CAP has been shown to control actin dynamics; second, an editorial decision on whether the resulting paper is more appropriate for *eLife* or a parasitology journal. Either answer is fine.

[Editors’ note: what now follows is the decision letter after the authors submitted for further consideration.]

Thank you for resubmitting your work entitled "Differential requirements for cyclase-associated protein (CAP) in actin-dependent processes of *Toxoplasma gondii*" for further consideration at *eLife*. Your revised article has been favorably evaluated by Anna Akhmanova as the Senior Editor, a Reviewing Editor, and three reviewers.

The manuscript has been improved but there are some remaining issues that need to be addressed before acceptance, as outlined below:

As you will see the two of the three reviewers now recommend publication. I would thus invite you to address the comments, which should only take a few days and resubmit with a point to point response.

Thanks for sending your manuscript to *eLife* and for revising it according to the suggestions of the reviewers.

Reviewer #1:

In their revision the authors provide additional data regarding the function of CAP in regulating F-actin dynamics (rather accumulation) in the parasite. Briefly, the authors demonstrate that depletion of CAP results in loss of the previously identified F-actin polymerization centre, close to the Golgi/apicoplast), while the intravacuolar network is still formed. Intriguingly, loss of F-actin polymerization close to the apicoplast does not result in a defect in apicoplast inheritance, as observed in case of a mutant for Formin-2 or the actin depolymerization factor ADF.

Based on these findings the authors speculate that F-actin polymerisation at/close to the apicoplast is not solely responsible for segregation of this organelle. However a detailed analysis of F-actin dynamics is still missing and the authors reach this conclusion based on differences in F-actin accumulation.

Therefore, another (potentially more plausible) interpretation is that in absence of FH-2 F-actin nucleation at the apicoplast is completely abrogated, while depletion of CAP leads to a significant reduction, but not abolishment of F-actin dynamics. This slower dynamics would still result in a thicker intravacuolar network, but could still be sufficient for segregation of the apicoplast. While it would be interesting to analyse and compare F-actin dynamics in these mutants in more detail, within this study, it is in my opinion sufficient to weaken the discussion accordingly (see below).

A major concern is the invasion assay employing Cb-EmeraldFP parasites. While the authors show convincingly that the junction is formed using the junctional marker Ron4 (Figure 8H), this is expected, since depletion of actin (or other components of the acto-myosin system) has no influence on junction formation (Whitelaw et al., 2017; Egarter et al., 2014).

However, the images provided in Figure 8G, indicating F-actin accumulation at the junction, are not convincing and the signal could well be due to a compression effect at the junction during invasion. Also, potential differences in F-actin accumulation at the junction need to be compared with WT parasites. Since this would be a longer endeavour and would not significantly add to the current study, this reviewer suggests to remove this set of data until a more convincing analysis has been performed.

In summary the authors addressed my major concerns. The study is a thorough phenotypic characterisation and the additional data presented demonstrate that CAP has indeed a function in F-actin dynamics and will open the door for future experiments.

Reviewer #2:

Speaking (again) as a worker on actin genetics, with a side interest in apicomplexan actin.

This paper now justifies the way it positions itself as an actin paper. Figure 8 in particular is clear and adds another chapter to the huge range of functions that actin seems to mediate. CAP is a bit of a mystery as it is; it is even more interesting as an unusually conserved molecule.

It is still overwhelmingly a parasite paper, and makes a demanding read for someone who's not soaked in the field; and I'll again have to leave it to the other reviewers to comment on whether the parasitology is sound.

Reviewer #3:

In this work, Hunt et al. analyzed the function of the *Toxoplasma* cyclase-associated protein (CAP). The authors discovered the role of CAP in regulating rosette formation as well as the surprising uncoupling of rosette formation and synchronicity of parasite replication. The experiments were well-executed and conclusions well-reasoned. The correlative FRAP and FIB-SEM experiments are not only technically challenging, but also very informative. In addition, the construction of the second generation DiCre parasite line is going to be a very useful tool for the field. The authors have fully addressed the comments from the previous review. I do not have any additional concerns.

---

## [Author Response]

[Editors’ note: the author responses to the first round of peer review follow.]

Reviewer #1:In the study Hunt et al., the authors perform a careful phenotypic characterisation of parasite mutants for the actin capping protein of the apicomplexan parasite Toxoplasma gondii, TgCAP. Using state of the art and cutting edge technologies the authors demonstrate that CAP is alternatively translated, localised and that it's deletion causes effects in parasite growth, probably caused by defects in host cell invasion and egress. Furthermore, deletion of CAP causes alteration of the residual body, a structure connecting parasites within the parasitophorous vacuole and consequently to inefficient parasite connectivity and material transfer. In contrast, other actin dependent processes, such as apicoplast segregation or dense granule motility appear to be not or only minimally affected in absence of CAP, leading the authors to speculate that CAP is only required in the regulation of certain actin-dependent processes.The authors present a tour-de-force to analyse the phenotype caused by deletion of cap focussing on processes, previously described to be dependent on parasite actin and/or myosins. The variety of techniques, from FIB SEM, SEM, FRAP, mouse challenge, FACs, western blot and live 3D motility assay, gives a great overview of the phenotypic consequences caused by deletion of CAP. While this characterisation is very well performed, it does not directly demonstrate a role of CAP in F-Actin dynamics in the parasite. It is now possible to measure and visualise F-actin dynamics in apicomplexan parasites, which allowed to demonstrate the role of F-actin in material transfer (Periz et al., 2017) and the independent functions of apicomplexan Formins in F-actin nucleation (see Tosetti et al., 2019). While the interpretation of the authors is certainly likely, no direct functional data regarding the role of CAP for the regulation of F-actin dynamics is provided and the conclusions are drawn based on the phenotypes obtained for other F-actin regulatory proteins. Therefore, in their resubmission, the authors should directly analyse the role of CAP in F-actin dynamics.1) This study is about a major actin modulating protein and does not look at actin dynamics directly. However, this is now possible using the F-actin nanobody and should be done. It would be of great interest, how short CAP, long CAP and ∆CAP influence actin dynamics during intracellular replication, egress, gliding motility and invasion.

To monitor F-actin using the chromobody we have deleted CAP in the chromobody-expressing line and complemented a clone with either WT, short or long CAP.

These experiments, now shown in Figure 8D and E, show a loss of the actin polymerisation centre in intracellular ∆CAP parasites that is rescued by short CAP, but only partially by long CAP. A second striking phenotype observed in CAP deficient intracellular parasites is the presence of extensive chromobody positive structures throughout the disorganised rosette, indicating actin filamentation. This is rescued by WT and short CAP, but not/only partially by long CAP in a rosette-dependent manner.

We also attempted to perform live invasion and egress videos with these parasite lines. While we could observe egress, and a modest amount of invasion events, with the WT CAP line, the combination of motility and egress phenotypes meant we only observed, despite many hours of microscopy time, only 3 successful invasion events for ∆CAP parasites. While these appeared to be slower than WT CAP, none were events captured from the very onset of invasion so timings could not be reported. We do recognise that analysis of the ∆CAP could contribute to our understanding of *Toxoplasma* invasion, especially in light of recent work which shows parasite nucleus entry into the host cell is a limiting step for which actin is an important player (Rosario et al., 2019 pre-print), however, we believe that doing so carefully for the mutant strains (and maybe more mutants as suggested by reviewer 2) are a huge undertaking and better suited for subsequent studies.

We were able to identify actin ring formation and presence of a MJ during invasion of ∆CAP parasites (Figure 8G). While the moving junction still appeared to form, CAP did not localise to the moving junction, suggesting that CAP is not required for moving junction formation. However, we cannot exclude that it is the turnover of actin at the moving junction that is important for successful invasion. Again, a very careful analysis of actin in CAP deletions and its variants during invasion would be required.

2) The authors did not look at the dynamics of CAP during these processes, which would be relatively straight forward, given that parasites expressing GFP-tagged versions are already available.Especially the dynamics of apically localized CAP would be interesting to analyse in more detail, since it might suggest an association with Formin1, whereas the cytoplasmic CAP might be more associated with Formin2/3. This might also reflect the different effects in the complementation assays.This should at least be discussed in more detail in a resubmission.

We have not tried to image the GFP-tagged CAP version because the CAP-GFP line still has endogenous CAP so it is an over-expression. We know from *Plasmodium* that over-expressing CAP can have negative effects not seen in a KO (Sato et al., 2016). So it might not be worth doing experiments on a line that has not been carefully assessed for such effects.

However, to still answer this question we decided to use the HA-tagged CAP parasites (∆CAP, ∆CAP^CAP^, ∆CAP^CAP^ and ∆CAP^longCAP^) and co-stain with RON4, that marks the moving junction. These assays show that CAP does not appear to localise to the moving junction and, in the absence of CAP, the moving junction still forms as shown by RON4 and the actin chromobody in invading parasites (Figure 8G and H).

The potential interplay of CAP with the Formins is now discussed further given that we see only partially overlapping phenotypes between CAP and these actin regulators.

3) The rosetting phenotype as well as the RTORP are very interesting findings. It is great to do CLEM on FRAP / FIB imaging, which to our knowledge is the first time this has been done in Toxoplasma. It is technically extremely challenging, as is indicated by the lack of statistical analysis for the FIB data. However, the discussion of the part of the analysis is a bit biased. One could immediately question if the correct time scale was analyzed for the FRAP analysis in those parasites lacking a central residual body. A long and tubular cytoplasmic bridge connecting two parasites might result in much longer diffusion times for cytoplasmic proteins.

The reviewer is correct that these experiments are very time consuming and technically challenging and we were aware that we could not exclude slow transfer of proteins between parasites, which we have discussed in the manuscript (see quote below from our Discussion). We have previously attempted to visualize photobleached parasites over longer periods of time, however, we see substantial drift of parasites, which makes quantification on individual ROIs very difficult. We therefore decided not to include these data in the original manuscript. We have not been able to improve the drift issues. This prevents analysis of slower diffusion leading us to report only rapid protein transfer differences.

From our Discussion:

“This connection, although not facilitating rapid transfer of proteins between parasites that are further apart, could allow for slow or minimal transfer of proteins which is sufficient to synchronise divisions. An alternative hypothesis could be that metabolites, not proteins, are required to synchronise divisions, and their diffusion through the decentralised residual body is quicker.”

Also, as actin was not visualized, more prominent actin filaments (caused by deletion of CAP) within the tubular residual body might delay diffusion of mCherry.

Agreed but very hard to prove definitively. We thought hard about how to solve this and the only way we think this was doable was to perform CLEM on FRAP-ed parasites where we observe strong actin filaments between some parasites and prove that these are within the tubules. However, as controls we would need tubules that are of equal length with a similar tubular content. As explained in the text and further below, levels of chromobody in the tubules appears to correlate with, and may be affecting, the tubular content. At this stage we think it is hard to be certain what reduces diffusion rates. It could be due to actin within the tubules, the presence/absence of the ER-like structures in the tubules, the distance between the parasites or another factor.

Similar results as the FIB SEM could have been generated in a quantifiable manner by expressing the actin-binding chromobody (depending on the localization within the various CAP mutants) or a strong fluorescent reporter expressed at the plasma membrane.

This was considered: however, the suggested experiment assumes that there would be chromobody signal in all of the decentralised residual body connections and is therefore a readout for cell connectivity, which may not be accurate.

We decided to invest more time to answer this important question first by performing CLEM on a chromobody expressing strain in which CAP has been deleted. We show, in Figure 9 and Video 10, that the extensive actin filaments in the vacuole are found within the decentralised residual body tubules that connect the parasites, however, the signal is highly heterogenous and appears to affect the overall tubular content. While this may explain how parasites remain connected throughout divisions, correlating FRAP signals to these highly heterogenous tubular connections would be challenging and require a substantial amount of CLEM experiments.

However, we believe that the quantification of FRAP and synchronicity of division, together with the CLEM data we provide lends sufficient data to support our claims that the CAP mutants are still connected and that this maintains high rates of synchronous division. However, in the future it would be interesting to see whether the few parasites that do not divide in synchronicity (~10%) are indeed not connected, which is what we would hypothesise.

Reviewer #2:Actin plays a critical role in the lytic cycle of apicomplexan parasites, and is known to be important for motility, organelle inheritance, and cell-cell communication. This manuscript explores the role of the putative actin regulator cyclase-associated protein (TgCAP) in a variety of such actin-dependent processes. A methodological improvement introduced in this manuscript is the stable DiCre construct, which retains its excision capacity over long periods of time.The authors provide evidence for two protein isoforms of TgCAP that arise from two translation start sites and result in a cytosolic short isoform and a long isoform that is apically localized through palmitoylation sties not present in the short isoform. While effects on a number of different processes are measured for TgCAP mutants, the authors never directly observe how or whether actin turnover is affected, nor do they explore how the distinct localizations contribute to the role of each isoform.Previous studies (Frénal et al., 2017, Tosetti et al., 2019) have observed asynchronous division in parasites with defects in intravacuolar cell-cell communication, upon disruption of F-actin and a specific myosin. One contrasting finding in the current manuscript is that loss of TgCAP can decouple these two phenotypes: reporter proteins do not readily diffuse between parasites in the entire vacuole, yet cell division remains synchronous. As detailed below, more work is needed to determine whether the TgCAP phenotype is truly distinct from the previously reported phenotypes. However, based on the previous reports involving actin and myosin in intravacuolar contact/communication, and the lack of mechanistic information regarding TgCAP, the current study does not significantly advance our understanding of the role of actin in the Toxoplasma lytic cycle.1) This manuscript focuses on phenotypic assessment of processes for which actin is important, but there are no direct observations of the effects of TgCAP on actin turnover, as is suggested in the title. This could be addressed by using modified actin-chromobodies, which have been employed to study actin dynamics in Toxoplasma (Periz et al., 2017), to visualize filamentous actin throughout the cells and networks of cells in the various TgCAP mutants.

See response to reviewer 1, concern 1.

2) In all the phenotypic assays, the short isoform of TgCAP is able to complement the knockout as well as or better than the long isoform of TgCAP. While the authors provide compelling evidence that the two isoforms have distinct cellular localizations, there is no clear evidence that the localization is playing a critical role. One explanation for the more severe defects in complementation observed with the long isoform is that it is expressed at substantially lower levels relative to the short isoform (Figure 3F), and that overall levels of TgCAP are critical for its function. A prediction of this would be that a palmitoylation mutant (C6,8A) that could only be expressed as the long isoform (M37L mutant) would be unable to rescue the phenotypes as well as the short isoform.

LongCAP expression contributes ~25% of all CAP in the cell (Figure 3E), yet it can rescue most phenotypes to WT levels, apart from rosetting, the competition assay and the juxtanuclear actin polymerisation centre. If it was only the levels that make a difference, we would expect all phenotypes be affected in a similar way. In addition, the shortCAP isoform, which does not display reduced levels of expression, shows reduced growth in the competition assay, arguing for some localisation dependent phenotypes for the two isoforms. A more detailed study using mutants as proposed by the reviewer would surely be interesting, but we think better suited in a subsequent study that carefully investigates the isoforms in processes such as invasion and in combination with other actin regulators. We have added to the Discussion that the levels of expression may be a contributing factor to the phenotypes although we believe they are insufficient to explain all phenotypic differences.

3) An important finding in this manuscript is that while TgCAP mutants have defects in protein diffusion between all parasites within a vacuole, they are still able to divide synchronously. Parasite division synchronicity assays presented in this manuscript are performed at 20 hours post infection. Frénal et al., showed increased severity in defects in synchronized division of myosin knockouts that occurred from 24 to 30 hours post infection. To ensure that the lack of synchronization defects observed in this manuscript is not simply due to earlier measurements, the authors should measure whether TgCAP mutants are still able to maintain synchronous divisions at these slightly later timepoints when there might be differences compared to wildtype.

This is a good suggestion and we have now also looked at 24 h and 30 h timepoints and indeed observe a slight, but statistically significant increase of asynchrony of daughter cell division at these later timepoints (Figure 6B). However, this reduction in synchronous division is small compared to that previously observed in Frénal et al., where only 30-40% divide synchronously for MyoI and MyoJ KO at the 30 h mark compared to ~88% in our analysis, despite complete lack of rosetting in all of these strains. Furthermore, Tosetti et al. reported that Formin 3 had only ~35% synchronous division but the time post-infection was not mentioned. In our study we propose that synchronicity may be achieved by parasites connected through the decentralised residual body. Lack of RTORP and rosetting does not predict connectivity as cells with a defect in either can still divide predominantly in synchronicity.

4) If levels of TgCAP are important for function, it is important to understand how the levels in the complemented strains compare to wildtype cells. This can be measured using the TgCAP antibody presented in this manuscript.

This was an important suggestion and we now provide this data in the manuscript, for the RH line (Figure 3E) as well as for the chromobody expressing line (Figure 8C). In both these instances, we see levels of CAP expression in the ∆CAP^CAP^ line comparable to WT parasites. We do observe a reduction of longCAP compared to WT parasites though, which is now mentioned in the text. However, the longCAP isoform can rescue the majority of ∆CAP phenotypes to WT levels, indicating that precise level of expression is not the main factor in longCAP function.

Reviewer #3:You've asked me to read this manuscript from the point of view of an actin biologist. And I'm afraid from that point of view this manuscript is not very enlightening – while it confirms that CAP is physiologically important, it can't pin down a function for the long form of CAP, makes no comment on what the CAP is doing to the actin, and all the data presented describe the behaviour of CAP alone in some detail but nothing of actin or actin regulatory proteins.From a reviewer's and editor's point of view, this leads to three conclusions:a) I'm really not competent to say whether the advances in apicomplexan biology are significant or not. They seem to be important, but I defer to reviewers with expertise in this area.b) However, the authors' attempts at justifying the results in terms of actin biology are not supported. There is no evidence that the CAP is having any effect on Toxoplasma's actin turnover. It should therefore be removed from the title. The evidence is only inferred from other systems. Additionally, since the stability of parasite actin is completely different (cf the excellent PNAS paper from Raunser's lab) even this inference is not trustworthy.

See response to reviewer 1, concern 1. We have now generated all CAP variants in the chromobody lines and show that CAP KO leads to loss of an actin polymerisation centre and aberrant chromobody filaments in the vacuole. We believe that this now lends further support for our conclusions. We have replaced the word “turnover” to “actin-dependent processes” in the title.

Similarly, the Discussion needs to be rewritten. It is not true that "our results strongly support an actin regulatory role for CAP in Toxoplasma". The results strongly support the biological importance of Toxoplasma CAP; the correlation with the biological importance of actin is very patchy. Again, the actin regulatory role is inferred from other work rather than shown here. The least indirect connection with actin is made via dense granule transport; however the MBoC paper that connects the two is already indirect (measuring loss of migration in the presence of strong actin inhibitors like cytochalasin and promoters like jasplakinolide, which of course cause secondary effects) – it is certainly not sound to conclude that a mutation that causes an opposite phenotype to actin inhibitors (dCAP mutant granules move fast) must solely be acting through actin dynamics.

The reviewer was correct to say that this was by inference. We hope that the chromobody experiments provide further support to our claims. In addition to providing the chromobody data we have made changes throughout the text to not imply any direct roles of CAP in actin turnover.

Of note to the reviewer: In Aoife Heaslip’s MBoC paper (Heaslip et al., 2016) she used the conditional actin knockout line, along with a Myosin F tail mutant, to test for dense granule motility, following the initial experiments with drugs that affect actin.

c) if this sounds very negative about the paper as a whole, it's not meant to be. The data seem clear and important. I'm not sure whether they belong in a general biology journal like eLife – perhaps that's the reason why the actin angle has been pushed harder than the data support.

We centred the manuscript on actin, as CAP is a predicted actin regulator and we tried to put our results in context to its predicted function. Our study on CAP now adds an important piece to the puzzle of how actin binding proteins in *Toxoplasma* allow the spatial regulation of actin. *Toxoplasma* with its reduced complexity of actin regulators, genetic toolbox and fairly well described phenotypes may therefore be a formidable model organism to study the interplay of actin regulators in a eukaryotic cell.

Along with the parasite specific data presented we believe the technical advances including generating stable dimerised enzymes and CLEM data of connectivity between cells or cellular compartments, are of interest to the broad readership. Although they do not form the core of this manuscript we feel they add to the interest for a general biology audience.

I would recommend – first, recast this as a paper on apicomplexan biology, with the Discussion raising the interesting possibility (yet again) that parasite actin has a cryptic biology that needs to be understood, rather than stating that CAP has been shown to control actin dynamics; second, an editorial decision on whether the resulting paper is more appropriate for eLife or a parasitology journal. Either answer is fine.

After revising the manuscript and with the addition of the actin chromobody data and feedback from non-parasitology colleagues, we believe that the manuscript is no longer too centred on actin.

[Editors' note: the author responses to the re-review follow.]

The manuscript has been improved but there are some remaining issues that need to be addressed before acceptance, as outlined below:Reviewer #1:In their revision the authors provide additional data regarding the function of CAP in regulating F-actin dynamics (rather accumulation) in the parasite. Briefly, the authors demonstrate that depletion of CAP results in loss of the previously identified F-actin polymerization centre, close to the Golgi/apicoplast), while the intravacuolar network is still formed. Intriguingly, loss of F-actin polymerization close to the apicoplast does not result in a defect in apicoplast inheritance, as observed in case of a mutant for Formin-2 or the actin depolymerization factor ADF.Based on these findings the authors speculate that F-actin polymerisation at/close to the apicoplast is not solely responsible for segregation of this organelle. However a detailed analysis of F-actin dynamics is still missing and the authors reach this conclusion based on differences in F-actin accumulation.Therefore, another (potentially more plausible) interpretation is that in absence of FH-2 F-actin nucleation at the apicoplast is completely abrogated, while depletion of CAP leads to a significant reduction, but not abolishment of F-actin dynamics. This slower dynamics would still result in a thicker intravacuolar network, but could still be sufficient for segregation of the apicoplast.

We have added this interpretation to the Discussion:

“An explanation for efficient apicoplast inheritance in the ∆CAP line, despite loss of the visible actin polymerisation centre, could be that F-actin dynamics are not completely abrogated. It is a possibility that Formin 2-mediated actin nucleation is still occurring in this region and is sufficient to ensure apicoplast segregation.”

While it would be interesting to analyse and compare F-actin dynamics in these mutants in more detail, within this study, it is in my opinion sufficient to weaken the discussion accordingly (see below).A major concern is the invasion assay employing Cb-EmeraldFP parasites. While the authors show convincingly that the junction is formed using the junctional marker Ron4 (Figure 8H), this is expected, since depletion of actin (or other components of the acto-myosin system) has no influence on junction formation (Whitelaw et al., 2017; Egarter et al., 2014).However, the images provided in Figure 8G, indicating F-actin accumulation at the junction, are not convincing and the signal could well be due to a compression effect at the junction during invasion. Also, potential differences in F-actin accumulation at the junction need to be compared with WT parasites. Since this would be a longer endeavour and would not significantly add to the current study, this reviewer suggests to remove this set of data until a more convincing analysis has been performed.

We agree that this set of data does not significantly add to the current study and would require considerable further experimentation to provide reasonable insight. As such, Figure 8G and H have been removed, as well as associated text in the Results and Discussion.